# A previously undescribed scene-selective site is the key to encoding ego-motion in naturalistic environments

**Bryan Kennedy[1], Sarala N Malladi[1], Roger BH Tootell[1,2], Shahin Nasr[1,2]\***

[1]Athinoula A. Martinos Center for Biomedical Imaging, Massachusetts General Hospital, Charlestown, United States; [2]Department of Radiology, Harvard Medical School, Boston, United States

**Abstract** Current models of scene processing in the human brain include three scene-selective areas: the parahippocampal place area (or the temporal place areas), the restrosplenial cortex (or the medial place area), and the transverse occipital sulcus (or the occipital place area). Here, we challenged this model by showing that at least one other scene-selective site can also be detected within the human posterior intraparietal gyrus. Despite the smaller size of this site compared to the other scene-selective areas, the posterior intraparietal gyrus scene-selective (PIGS) site was detected consistently in a large pool of subjects (n = 59; 33 females). The reproducibility of this finding was tested based on multiple criteria, including comparing the results across sessions, utilizing different scanners (3T and 7T) and stimulus sets. Furthermore, we found that this site (but not the other three scene-selective areas) is significantly sensitive to ego-motion in scenes, thus distinguishing the role of PIGS in scene perception relative to other scene-selective areas. These results highlight the importance of including finer scale scene-selective sites in models of scene processing – a crucial step toward a more comprehensive understanding of how scenes are encoded under dynamic conditions.

**\*For correspondence:** shahin.nasr@mgh.harvard.edu

**Competing interest:** The authors declare that no competing interests exist.

**Sent for Review** 28 August 2023
**Preprint posted** 25 September 2023
**Reviewed preprint posted** 16 January 2024
**Reviewed preprint revised** 07 March 2024
**Version of Record published** 20 March 2024

## eLife assessment

In this article, the authors present a wealth of fMRI data at both 3T and 7T to identify a scene-selective region of the intraparietal gyrus ('PIGS') that appears to have some responsivity to characteristics of ego-motion. In a series of experiments, they delineate the anatomical location of PIGS and functionally differentiate it from nearby V6 and OPA. Evidence for these **important** findings is **solid**, but further investigations as to the role of this region in processing ego-motion will be needed to confirm this conclusion.

## Introduction

In human and non-human primates (NHPs), fMRI has been used for many decades to localize the cortical regions that are preferentially involved in scene perception (*Epstein and Kanwisher, 1998*; *Tsao et al., 2008*; *Rajimehr et al., 2009*; *Nasr et al., 2011*). Early studies focused mainly on larger activity sites that were more easily reproducible across sessions and individuals, ignoring smaller sites that were not detectable in all subjects and/or were not reproducible across scan sessions, based on the techniques available at that time. This led to relatively simple models of neuronal processing solely based on larger visual areas.

These models suggested three scene-selective areas within the human visual cortex, with possible homologs in NHPs (*Nasr et al., 2011*; *Kornblith et al., 2013*; *Li et al., 2022*). The human cortical

**Figure 1.** Distribution of scene-selective areas within the human visual cortex. Panel (**A**) shows the group-averaged (n = 14) response to 'scenes > faces' contrast (Experiment 1). Areas parahippocampal place area/temporal place area (PPA/TPA), restrosplenial cortex/medial place area (RSC/MPA), and transverse occipital sulcus/occipital place area (TOS/OPA) are localized within the temporal, medial and posterior-lateral brain surfaces, respectively. To show consistency with our previous reports (**Nasr et al., 2011**), data from individual subjects was largely smoothed (FWHM = 5 mm) and the group-averaged maps were generated based on fixed- rather than random-effects (see also **Figure 3**). The resultant map was thresholded at p<10^{-25} and overlaid on the common brain template (fsaverage). Panel (**B**) shows the activity map in one randomly selected subject (see also **Figure 2**), evoked in response to the same stimulus contrast as in panel (**A**) Here, the activity map was only minimally smoothed (FWHM = 2 mm). Consequently, multiple smaller scene-selective sites could be detected across the cortex, including posterior intraparietal gyrus scene-selective site (PIGS) (black arrow), located within the posterior intraparietal gyrus. Traditionally, these smaller activity patches are treated as noise in measurements and discarded. For ease in comparing the two panels, the individual's data was also overlaid on the fsaverage.

areas were originally named parahippocampal place area (PPA) (**Epstein and Kanwisher, 1998**), retro-splenial cortex (RSC) (**Maguire, 2001**), and transverse occipital sulcus (TOS) (**Grill-Spector, 2003**), based on the local anatomical landmarks. However, subsequent studies noticed the discrepancy between the location of these functionally defined areas and the anatomical landmarked, and instead named those regions temporal place area (TPA), medial place area (MPA), and occipital place area (OPA) (**Nasr et al., 2011**; **Dilks et al., 2013**; **Silson et al., 2016**).

The idea that scene-selective areas are limited to these three regions is based largely on group-averaged activity maps, generated after applying large surface/volume-based smoothing to the data from individual subjects. In such group-averaged data, originally based on fixed- rather than random-effects, thresholds tended to be high to reduce the impact of nuisance artifacts (**Nasr et al., 2011**). Thus, though well founded, this approach conceivably may not have identified smaller scene-selective areas (**Figure 1A**).

However, at the single-subject level, multiple smaller scene-selective sites can be detected outside these scene-selective areas, especially when drastic spatial smoothing is avoided (**Figure 1B**). This phenomenon is highlighted in a recent neuroimaging study in NHPs (**Li et al., 2022**) in which the authors took advantage of high-resolution neuroimaging techniques using implanted head coils. Their findings suggested that scene-selective areas are likely not limited to the three expected sites, and that other, smaller, scene-selective areas may also be detected across the brain. Still, the reliability in the detection of these smaller sites, their spatial consistency across large populations, and their specific role in scene perception that distinguishes them from the other scene-selective areas remain unclear.

Here, we used conventional (based on a 3T scanner) and high-resolution (based on a 7T scanner) fMRI to localize and study additional scene-selective site(s) that were detected outside PPA/TPA,

RSC/MPA, and TOS/OPA. We focused our efforts on the posterior portion of the intraparietal cortex mainly because multiple previous studies reported indirect evidence for scene and/or scene-related information processing within this region (*Lescroart and Gallant, 2019*; *Pitzalis et al., 2020*; *Sulpizio et al., 2020*; *Park et al., 2022*). Consistent with these studies, we found at least one additional scene-selective area within the posterior intraparietal gyrus, adjacent to the motion-selective area V6 (*Pitzalis et al., 2010*). This site was termed PIGS, reflecting its location (posterior intraparietal gyrus) and function (scene-selectivity). PIGS was detected consistently across individual subjects and populations and localized reliably across scan sessions. Besides its distinct location relative to two major anatomical landmarks (i.e., intraparietal sulcus [IPS] and parieto-occipital sulci [POS]) and the retinotopic visual areas (IPS0-4) that distinguishes it from other scene-selective areas (e.g., TOS/OPA and RSC/MPA), PIGS showed sensitivity to ego-motion within naturalistic visual scenes, a phenomenon not detectable in other scene-selective areas.

## Results

This study consists of seven experiments. Experiment 1 focused on localizing the scene-selective site (PIGS) within the posterior intraparietal region. Experiment 2 showed consistency in the spatial location of PIGS across sessions. Experiment 3 examined PIGS location relative to V6, an area involved in motion coherency and optic flow encoding, and also relative to the retinotopic visual areas IPS0-4. Experiment 4 showed that, despite its small size, PIGS is detectable in group-averaged maps in large populations. Experiment 5 showed that scenes and non-scene objects are differentiable from each other based on the evoked response evoked within PIGS. Experiment 6 tested the response in PIGS to ego-motion in scenes, yielding a result that differentiated PIGS from the other scene-selective regions. Finally, Experiment 7 showed that PIGS does not respond selectively to biological motion.

### Experiment 1: Small scene-selective sites are detectable within the posterior intraparietal gyrus

When the level of spatial smoothing is relatively low, scene-selective sites (other than PPA/TPA, TOS/OPA, and RSC/MPA) are detectable across the brain, especially within the posterior intraparietal gyrus

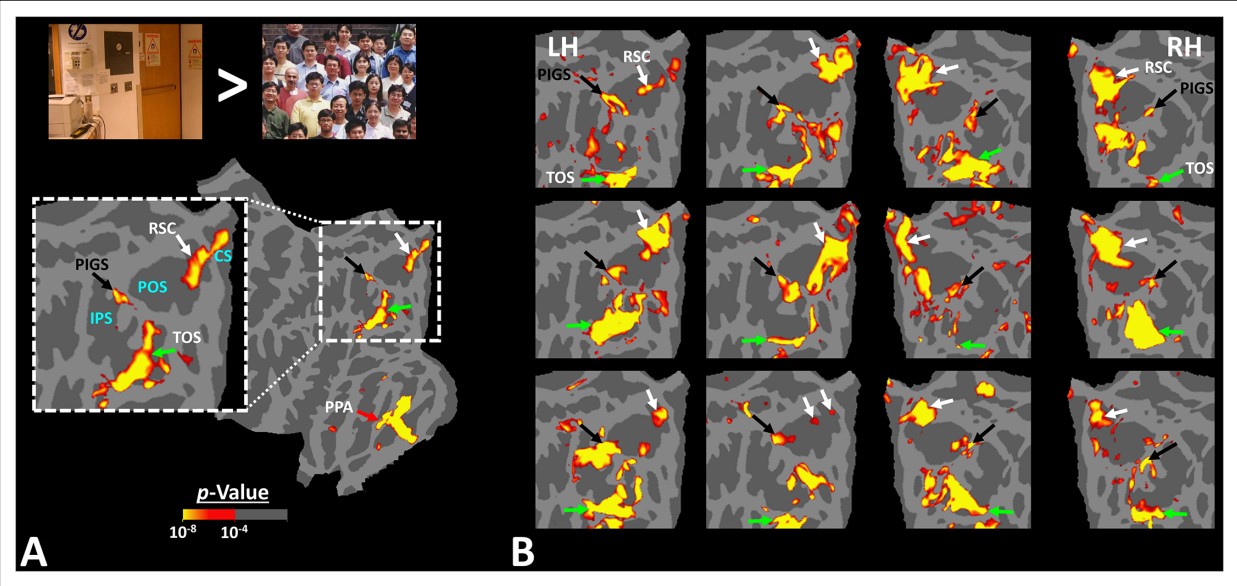

**Figure 2.** Activity evoked by 'scene > face' contrast in seven individual subjects, other than the one shown in *Figure 1*. Panel (**A**) shows the significance of evoked activity in the left hemisphere (LH) of one individual subject. The inset shows the enlarged activity map within the intraparietal region. The three scene-selective areas, along with area posterior intraparietal gyrus scene-selective site (PIGS), are indicated in the map with arrows. The location of the parieto-occipital sulcus (POS), the intraparietal sulcus (IPS), and the calcarine sulcus (CS) is also indicated in the inset. Panel (**B**) shows the result from six other individuals. In this panel, the first two columns show the activity within the LH, while the next two columns show the activity within the right hemisphere (RH) of the same subjects. In all subjects, PIGS (black arrow) is detectable bilaterally within the posterior portion of the intraparietal gyrus, near (but outside) the POS. All activity maps were overlaid on the fsaverage to highlight the consistency in PIGS location across the subjects.

(*Figure 1B*). To test the consistency in location of these scene-selective sites across individuals, 14 subjects were presented with scene and face stimuli while we collected their fMRI activity. Considering the expected small size of the scene-selective sites within the intraparietal region, we used limited signal smoothing in our analysis (FWHM = 2 mm; see 'Methods') to increase the chance of detecting these sites.

*Figure 2* shows the activity maps evoked by the 'scenes > faces' contrast in seven exemplar subjects. All activity maps were overlaid on a common brain template to clarify the consistency in the location of scene-selective sites across individuals. In all tested individuals, besides areas RSC/MPA and TOS/OPA, we detected at least one scene-selective site within the posterior portion of the intraparietal gyrus, close to (but outside) the POS. Accordingly, we named this site the posterior inter-parietal gyrus scene-selective site (PIGS).

When measured at the same threshold levels ($p<10^{-2}$), the relative size of PIGS was 73.86% ± 49.01% (mean ± SD) of RSC/MPA, 28.26% ± 15.67% of TOS/OPA, and 19.45% ± 8.43% of PPA/TPA. Considering the proximity of PIGS to the skull and head coil surface (*Figure 1*), the relatively small size of PIGS could not be ascribed to the lower signal-/contrast-to-noise ratio in that region.

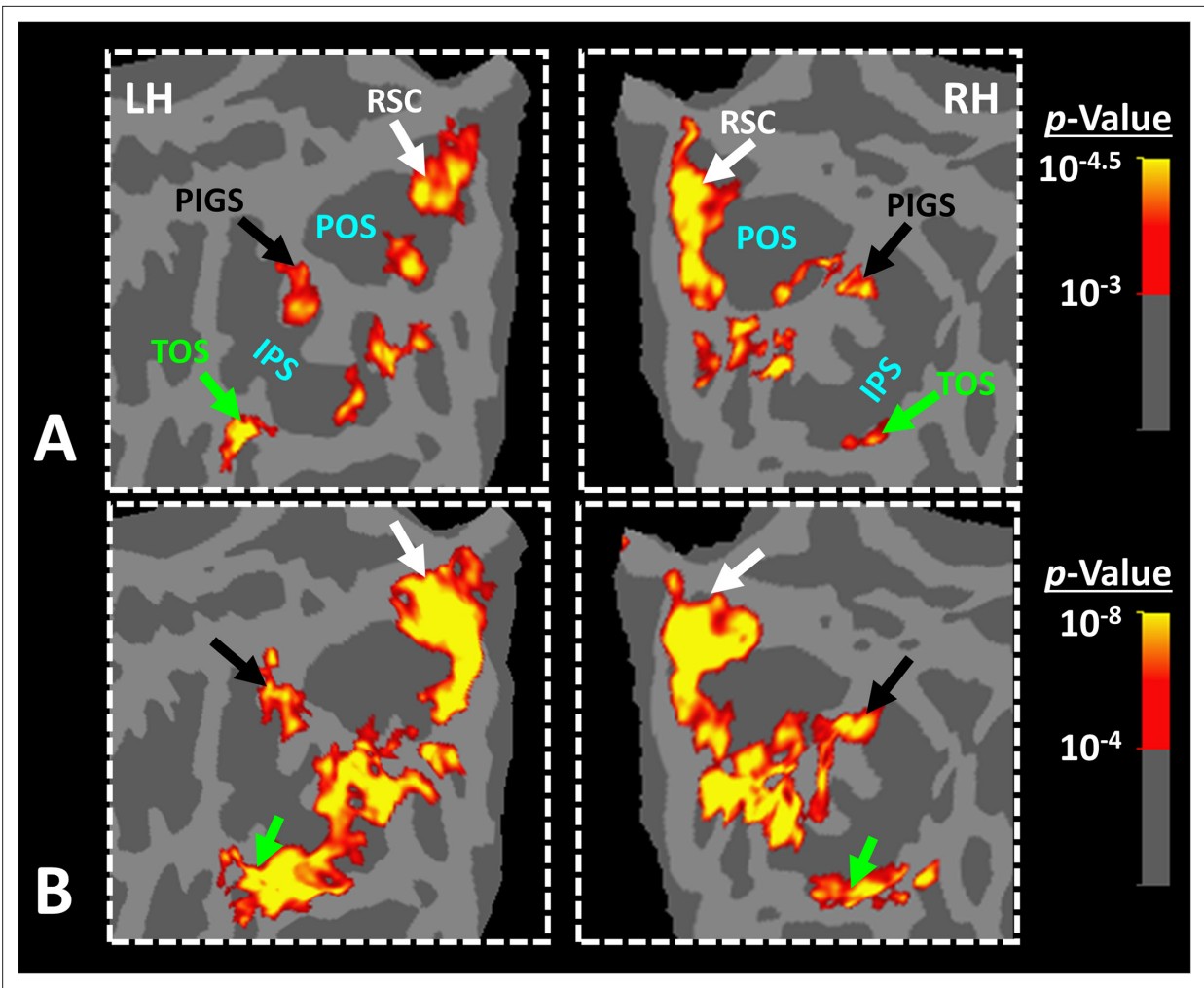

**Figure 3.** Posterior intraparietal gyrus scene-selective site (PIGS) was detected in group-averaged activity maps across two non-overlapping populations. Panel (**A**) shows the group-averaged activity, evoked within the intraparietal region of 14 subjects who participated in Experiment 1. Panel (**B**) shows the group-averaged activity, evoked within the intraparietal region of 31 subjects who participated in Experiment 4. Importantly, PIGS was evident in both groups bilaterally in the corresponding location (black arrows). Thus, despite its small size, this area was detectable even in the group-averaged activity maps based on large populations. Notably, in both panels, maps were generated based on random-effects, after correction for multiple comparisons. In both maps, the location of restrosplenial cortex/medial place area (RSC/MPA) and transverse occipital sulcus/occipital place area (TOS/OPA) are respectively indicated with white and green arrows.

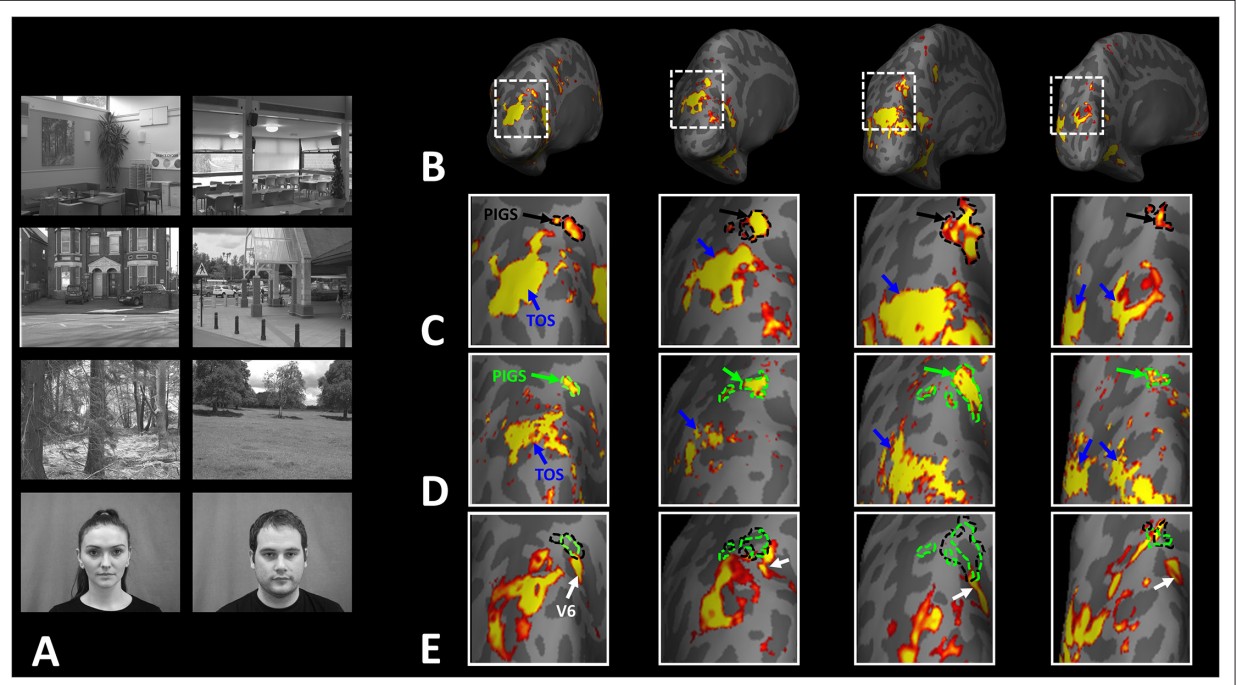

**Figure 4.** Posterior intraparietal gyrus scene-selective site (PIGS) was detected consistently across sessions. Panel (**A**) shows the stimuli used for localizing PIGS during 7T scans. Stimuli including indoor, manmade outdoor, and natural outdoor scenes and faces other than those used in Experiment 1. Panels (**B**) and (**C**) show the significance ($p<10^{-2}$) of activity evoked by 'scene > face' contrast in the 3T scans (Experiment 1), overlaid on subjects own reconstructed brain. Panel (**D**) shows the significance ($p<0.05$) of activity evoked by 'scene > face' contrast during 7T scans (Experiment 2). Despite the difference in scanners (3T vs. 7T) and stimuli, the location of PIGS remained mostly unchanged. Panel (**E**) shows the location of PIGS, measured in 3T (black dashed lines) and 7T (green dashed lines) relative to the location of area V6 (white arrow), localized functionally based on the response to 'optic-flow > random motion' (Experiment 3 a). In all subjects, the center of scene- and optic-flow-selective responses was adjacent, but not overlapping.

To better clarify the consistency of PIGS localization across subjects, we also generated group-averaged activity maps based on random-effects, and after correction for multiple comparisons. As demonstrated in *Figure 3A*, PIGS was also detectable in the group-averaged activity maps, in almost the same location as in the individual subject maps. Overall, these results suggest that, despite the relatively small size of this scene-selective site, PIGS is consistently detectable across subjects in the same cortical location.

## Experiment 2: PIGS reproducibility across scan sessions

To test the reproducibility of our results, four subjects were selected randomly from those who participated in Experiment 1. These subjects were scanned again (on a different day) using a 7T (rather than a 3T) scanner and a different set of scenes and faces (*Figure 4A*).

As demonstrated in *Figure 4*, despite utilizing a different scanner and a different set of stimuli, PIGS was still detectable in the same location (*Figure 4B–D*). Here again, PIGS was localized within the posterior portion of the intraparietal gyrus and close to the posterior lip of POS. Considering the higher contrast/signal-to-noise ratio of 7T (compared to 3T) scans, this result strongly suggested that the PIGS evidence was not simply a nuisance artifact in fMRI measurements.

## Experiment 3: Localization of areas PIGS vs. V6 and retinotopic visual areas

Posterior intraparietal cortex accommodates area V6, which is involved in motion coherency (optic-flow) encoding (*Pitzalis et al., 2010*). Recent studies have suggested that scene stimuli evoke a strong response within V6 (*Sulpizio et al., 2020*). Moreover, the intraparietal cortex accommodates multiple retinotopically organized visual areas (*Swisher et al., 2007*), including IPS0-4 that are believed to be involved in spatial attention control and higher-level object information processing (*Silver et al., 2005*; *Konen and Kastner, 2008*). Previous studies have suggested that the area TOS/OPA overlaps

with the retinotopic visual areas V3A/B and IPS0 (V7) (*Nasr et al., 2011*; *Silson et al., 2016*). In Experiment 3, we clarified the location of PIGS relative to these regions.

In Experiment 3a, we localized V6 in all subjects who participated in Experiment 1, based on visual presentation of random vs. radially moving dots (see 'Methods'). *Figure 4D* shows the co-localization of V6 and PIGS in four individual subjects. Consistent with previous studies (*Pitzalis et al., 2010*; *Pitzalis et al., 2015*), V6 was localized *within* the posterior portion of the POS without any overlap between its center and PIGS. To test the relative localization of these two regions at the group level, we generated probabilistic labels for PIGS and V6 (see 'Methods'). As demonstrated in *Figure 5*, the probabilistic label for PIGS was localized within the intraparietal gyrus and outside the POS (*Figure 5A*), while V6 was located within the POS (*Figure 5B*). We also did not find any overlap between area V6 and areas RSC/MPA and TOS/OPA (*Figure 5C*). Thus, despite the low threshold level used to generate these labels (probability >20%), the areas PIGS and V6 were located side-by-side (*Figure 5D*), without any overlapping between their centers.

In Experiment 3b, we scanned two subjects, randomly selected from those who had participated in Experiment 2, using a 7T scanner to map the borders of retinotopic visual areas (see 'Methods'). As demonstrated in *Figure 6*, in both subjects, PIGS was located adjacent to IPS3 and IPS4. In comparison, TOS/OPA was located more ventrally relative to PIGS, overlapping with areas V3A/B and IPS0 (V7). Considering these differences in the localization of PIGS vs. TOS/OPA, relative to the anatomical and functionally defined landmarks, our results further suggest that PIGS and TOS/OPA are two distinct visual areas.

## Experiment 4: PIGS localization in a larger population

The results of Experiments 1–3 suggest that PIGS can be localized consistently across individual subjects, and this area appears to be distinguishable from the adjacent area V6. However, considering the small size of this area, it appears necessary to test whether this area was detectable based on group averaging in a larger population. Accordingly, in Experiment 4 we scanned 31 individuals (other than those who participated in Experiments 1–3) while they were presented with the same stimuli as in Experiment 1 (*Figure 2*).

As demonstrated in *Figure 3B*, PIGS was also detectable in this new population in almost the same location as in Experiment 1. Specifically, PIGS was detected bilaterally within the posterior portion of the intraparietal gyrus, adjacent to the POS. We did not find a significant difference between the two populations in the size of PIGS when normalized either relative to the size of RSC/MPA (t(43) = 0.98, p=0.33), or TOS/OPA (t(43) = 0.26, p=0.80) or PPA/TPA (t(43) = 0.52, p=0.61). Thus, the location and relative size of PIGS appeared to remain unchanged across populations.

These results suggest that one may rely on the probabilistically generated labels to examine the evoked activity within PIGS. To test this hypothesis, we measured the level of scene-selective activity in PIGS, along with the areas TOS/OPA, RSC/MPA, and V6, using the probabilistic labels generated based on the results of Experiments 1 and 3a (see 'Methods' and *Figure 5*). As demonstrated in *Figure 7A and B*, results of this region of interest (ROI) analysis showed a significant scene-selective activity in PIGS (t(31) = 8.11, p<$10^{-8}$), TOS/OPA (t(31) = 7.91, p<$10^{-7}$), and RSC/MPA (t(31) = 9.11, p<$10^{-8}$). Importantly, despite the proximity of PIGS and V6, the level of scene-selective activity in PIGS was significantly higher than that in V6 (t(11) = 5.03, p<$10^{-4}$). Thus, it appears that the probabilistically generated ROIs can be used to examine PIGS response and differentiate it from adjacent areas such as V6 (see also Experiment 5).

## Experiment 5: Selective response to scenes compared to non-scene objects in PIGS

Thus far, we localized PIGs in multiple experiments by contrasting the response evoked by scenes vs. faces. In Experiments 5a and 5b, we examined whether PIGS also showed a selective response to scenes compared to objects (not just faces). In Experiment 5a, 12 individuals, other than those who participated in Experiments 1–3, were scanned while viewing pictures of scenes (other than those used to localize PIGS) and everyday objects (*Figure 8A*; see Methods).

As demonstrated in *Figure 8B and C* for one individual subject, 'scenes vs. objects' and 'scenes vs. faces' (Experiment 4) contrasts generated similar activity maps. Importantly, in both maps, PIGS was detectable in a consistent location adjacent to (but outside) the POS. Moreover, results of an

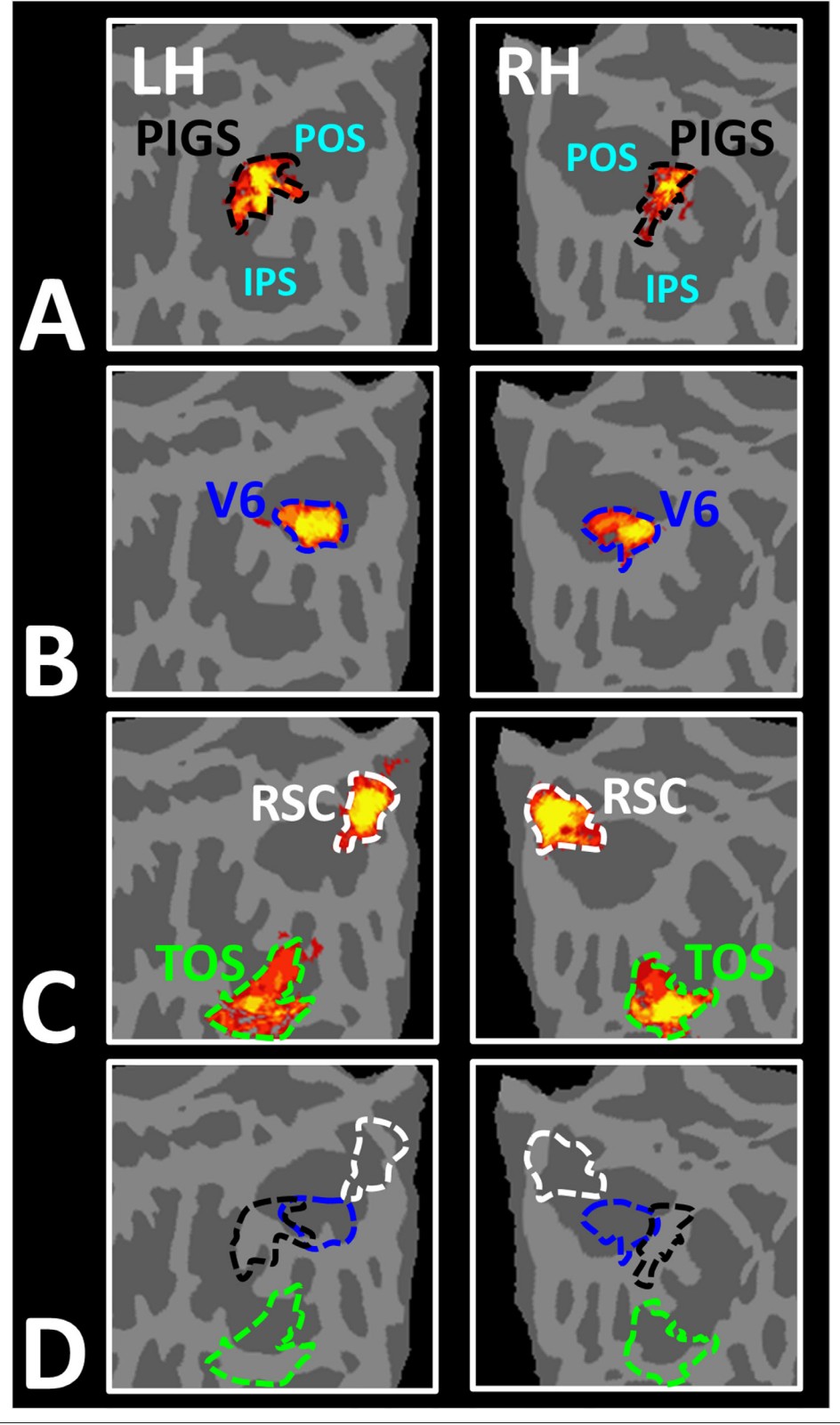

**Figure 5.** Area posterior intraparietal gyrus scene-selective site (PIGS) is located outside the parieto-occipital sulci (POS) and adjacent to the functionally localized area V6. Panels (**A**) and (**B**) show the probabilistic localization of areas PIGS and V6, respectively (see 'Methods'). Panel (**C**) shows the probabilistic localization of areas restrosplenial cortex/medial place area (RSC/MPA) and transverse occipital sulcus/occipital place area (TOS/OPA).

*Figure 5 continued on next page*

*Figure 5 continued*

All probability maps are thresholded at 20–50% (red-to-yellow) and overlaid on the fsaverage. Panel (**D**) shows the relative location of these sites. Consistent with the results from the individual maps (*Figure 4E*), PIGS and V6 were located adjacent to each other, such that V6 was located within the POS and PIGS was located outside the POS (within the intraparietal gyrus) with minimal overlap between the two regions.

ROI analysis, using the probabilistically generated labels based on the results of Experiments 1 and 3a, yielded significant scene-selective activity within PIGS (t(11) = 6.57, p<$10^{-4}$), RSC/MPA (t(12) = 11.00, p<$10^{-6}$), and TOS/OPA (t(12) = 6.26, p<$10^{-3}$) (*Figure 9A and B*). We also found that the level of scene-selective activity within PIGS is significantly higher than that in the adjacent area V6 (t(11) = 2.42, p=0.03). Thus, scenes and (non-face) objects are differentiable from each other, based on the activity evoked within PIGS.

In Experiment 5b, 15 individuals (other than those who participated in Experiments 1 and 5a), were scanned while viewing a new set of stimuli that included pictures of scenes, faces, everyday objects, and scrambled objects (*Figure 8D*). In contrast to Experiment 5a in which the number of objects

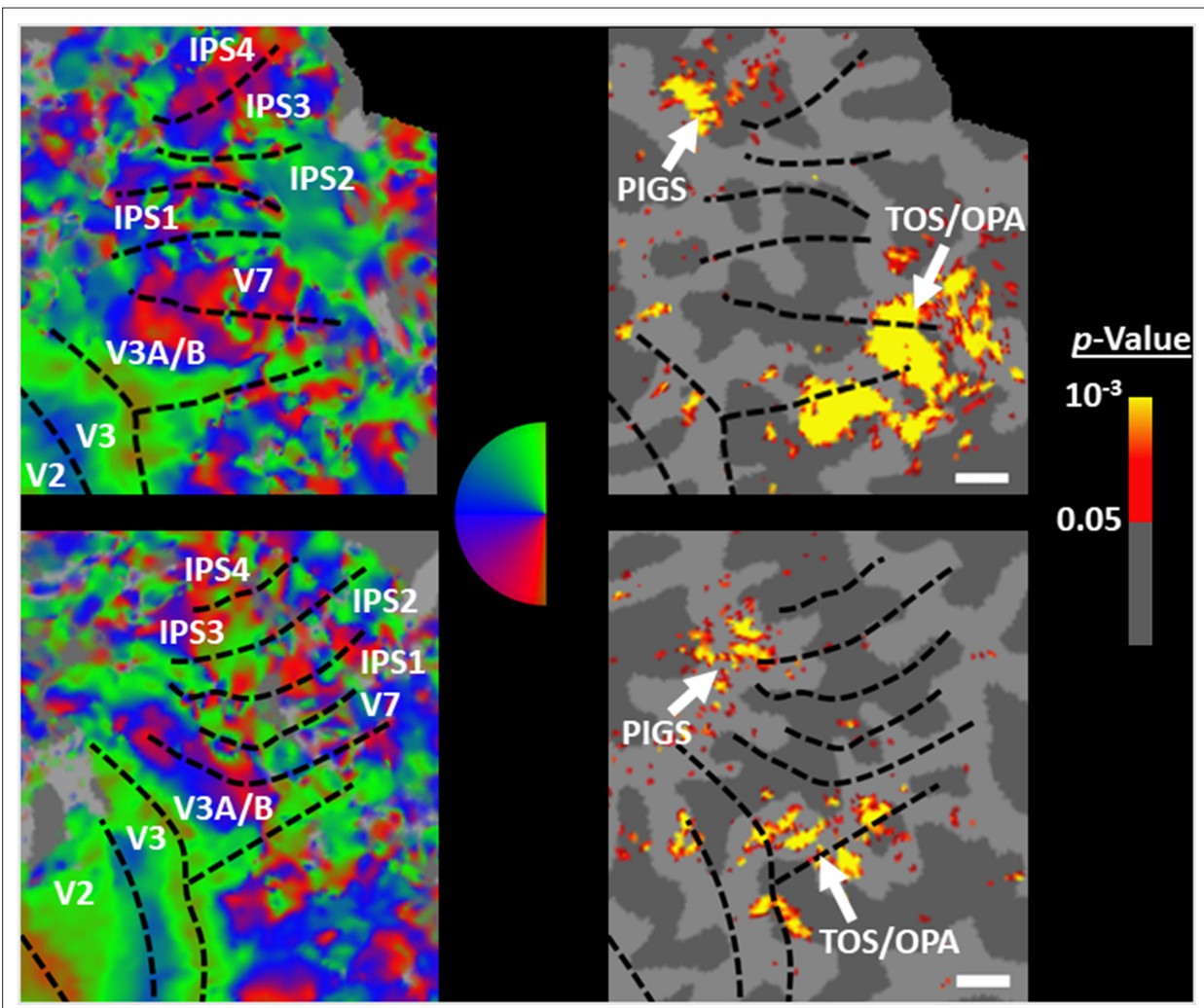

**Figure 6.** Localization of posterior intraparietal gyrus scene-selective site (PIGS) and transverse occipital sulcus/occipital place area (TOS/OPA) relative to the retinotopic visual areas in the right hemisphere of two subjects. The right and left columns show respectively the polar angle and scene > face response mapping, collected in a 7T scanner on two different days. In both subjects, PIGS was located close to areas IPS3-4. Area TOS/OPA overlapped with areas V3A/B and IPS0 (V7). The borders of visual areas (defined based on the polar angle mapping) are indicated by dashed black lines. Notably, for both subjects, maps were overlaid on their own reconstructed flattened cortex. No activity smoothing was applied to the collected data (i.e., FWHM = 0; see 'Methods'). Similar results were also found in the opposite hemispheres (not shown here). On the right column, the scale bars indicate 1 cm.

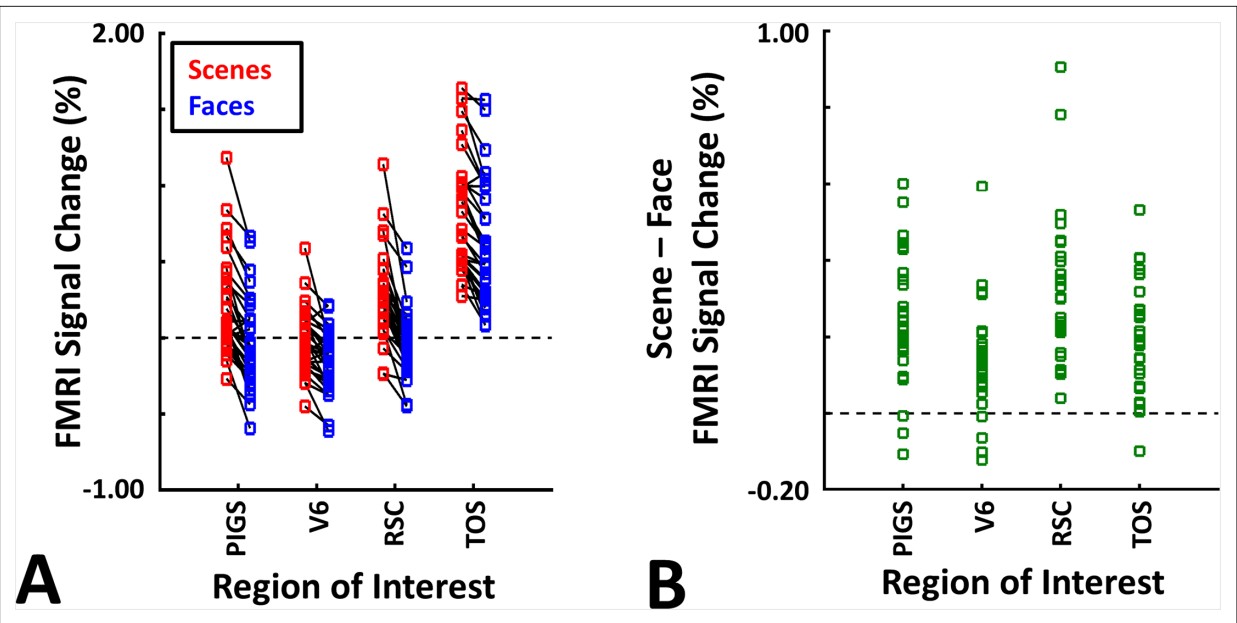

**Figure 7.** Probabilistically generated labels can be used to measure posterior intraparietal gyrus scene-selective site (PIGS) response in different experimental conditions. Panel (**A**) shows the activity evoked by scenes and faces, across PIGS, V6, restrosplenial cortex/medial place area (RSC/MPA), and transverse occipital sulcus/occipital place area (TOS/OPA), all of them localized based on probabilistically generated labels based on a different group of subjects. Panel (**B**) shows the level of scene-selective activity, measured as 'scene – face', within these regions. Despite the small size of PIGS, the probabilistic label could detect the scene-selective activity within this area and the level of this activity was significantly higher than the adjacent area V6. In all panels, each dot represents the activity measured in one subject.

within each image could vary, here, each image contained only one object (see 'Methods'). Despite this change, contrasting the response to scene vs. non-scene images (averaged over objects, scrambled objects, and faces) evoked a similar activity pattern, as scene vs. faces (*Figure 8E and F*). Moreover, the ROI analysis yielded a significant scene-selective activity within PIGS (t(14) = 2.37, p=0.03), RSC/MPA (t(14) = 10.33, p<10$^{-7}$), and TOS/OPA (t(14) = 4.79, p<10$^{-3}$) (*Figure 9*). Here again, the level of scene-selective activity within PIGS was higher than V6 (t(14) = 2.27, p=0.04). Together, results of Experiments 1–5 suggest that PIGS responds selectively to a wide range of scenes compared to non-scene objects, and that the level of this activity is higher than in the adjacent area V6.

## Experiment 6: PIGS response to ego-motion

Experiments 1–5 clarified the location of PIGS and its general functional selectivity for scenes. However, a more specific role of this area in scene perception remains undefined. Experiment 6 tested the hypothesis that area PIGS is involved in encoding ego-motion within scenes. This hypothesis was motivated by the fact that PIGS is located adjacent to V6 (*Figure 5D*), an area involved in encoding optic flow. Other studies have also suggested that ego-motion may influence the scene-selective activity within this region, without clarifying whether this activity was centered either within or outside V6 (*Pitzalis et al., 2020*; *Sulpizio et al., 2020*).

Twelve individuals, from those who participated in Experiment 1, took part in this experiment (see 'Methods'). These subjects were presented with coherently changing scene stimuli that implied ego-motion across different outdoor trails (*Figure 10*). In separate blocks, they were also presented with incoherently changing scenes and faces. *Figure 9* shows the group-averaged scene-selective activity, evoked by coherently (*Figure 11A*) and incoherently changing scene stimuli (*Figure 11B*). Consistent with our hypothesis, PIGS showed a significantly stronger response (bilaterally) to coherently (compared to incoherently) changing scenes that implied ego-motion (*Figure 11C*). However, the level of activity within RSC/MPA and TOS/OPA did not change significantly between these two conditions.

Consistent with the group-averaged activity maps, results of an ROI analysis (*Figure 12*) yielded a significantly stronger response to coherently (vs. incoherently) changing scenes in PIGS (t(11) = 5.97,

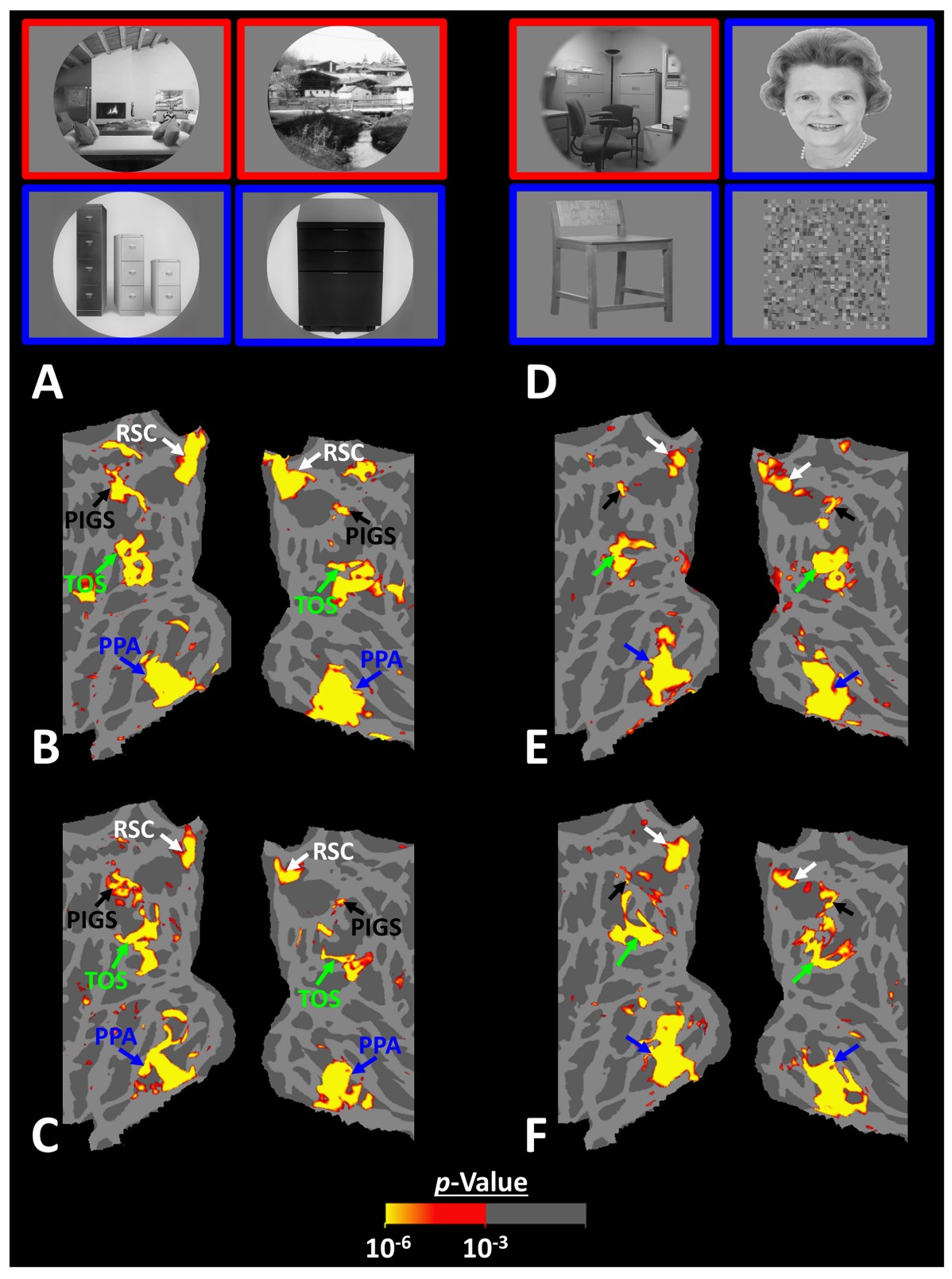

**Figure 8.** Posterior intraparietal gyrus scene-selective site (PIGS) could also be detected based on the 'scene > object' contrast. Panels (**A**) and (**D**) show the stimuli used in Experiments 5a and 5b respectively. Panels (**B**) and (**E**) show the activity maps evoked by 'scene > object' contrast in two different individuals who participated in Experiments 5a and 5b. Panels (**C**) and (**F**) show the activity maps evoked by a different set of scenes and faces (used in Experiments 1 and 4) in the same individuals. The location of PIGS, as indicated by the blacks, remained unchanged between the two maps.

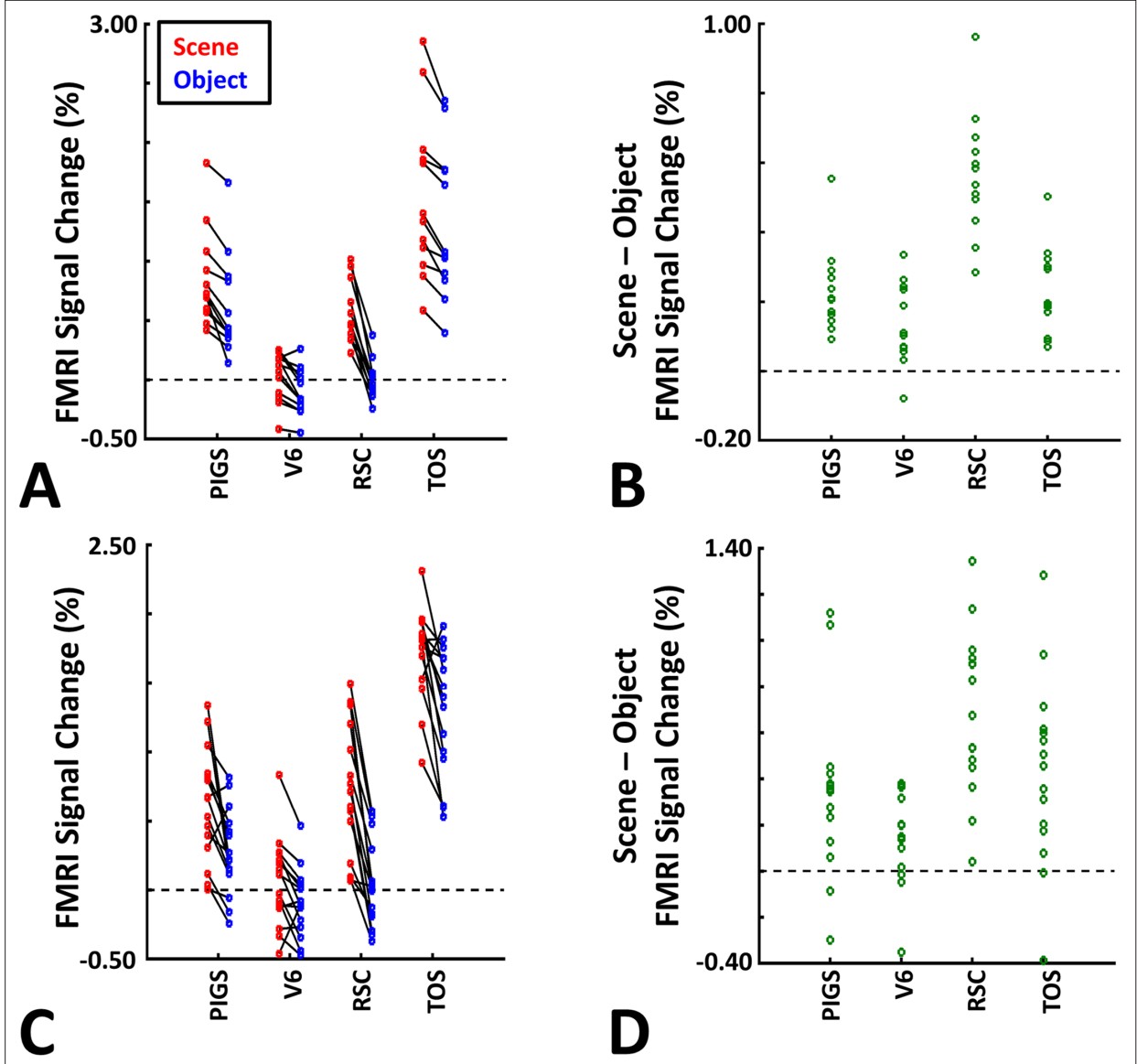

**Figure 9.** The application of probabilistically generated labels to measure the posterior intraparietal gyrus scene-selective site (PIGS) response to 'scene vs. object' stimuli. Panels (**A**) and (**C**) show the activity evoked by scenes and objects in Experiments 5a and 5b, respectively. Panels (**B**) and (**D**) show the level of scene-selective activity within the regions of interest. As in Experiment 4, the probabilistic label detected the scene-selective activity within PIGS and the level of this activity was significantly higher than the adjacent area V6. Other details are similar to *Figure 7*.

$p<10^{-4}$) but not in RSC/MPA (t(11) = 0.12, p=0.90) and TOS/OPA (t(11) = 0.48, p=0.64). Interestingly, area PPA/TPA showed a stronger response to incoherently (compared to coherently) changing scenes (t(11) = 3.48, p<0.01). To better clarify the difference between scene-selective areas, we repeated this test by applying a one-way repeated measures ANOVA to the differential response to 'coherently vs. incoherently changing scenes', measured across these four scene-selective areas. This test yielded a significant effect of area on the evoked differential activity (F(3, 11) = 53.89, $p<10^{-10}$). Post hoc analysis, with Bonferroni correction, showed that the level of differential activity evoked by 'coherently vs. incoherently changing scenes' was significantly higher within PIGS than all other scene-selective areas ($p<10^{-6}$). These results suggest a distinctive role for area PIGS in ego-motion encoding, which differentiates it from the other scene-selective areas. The absence of activity modulation in the other scene-selective areas also ruled out the possibility that the activity increase in PIGS was simply due to attentional modulation during coherently vs. incoherently changing scenes (see 'Discussion').

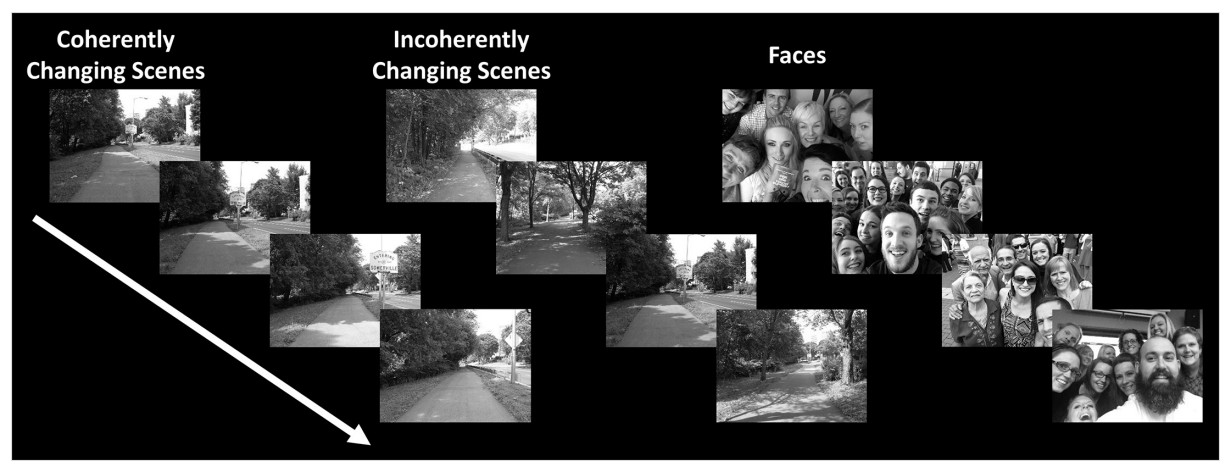

**Figure 10.** Example of stimuli used in Experiment 6. Coherently changing scenes implied ego-motion as if the observer was jogging through a trail. Incoherently changing scenes consisted of the same scene images as the coherently changing scenes but presented in a pseudo-random order. Face stimuli consisted of a mosaic of faces. These stimuli were different than those used in the previous experiments.

In addition to PIGS, we also found a significantly stronger response to coherently (rather than incoherently) changing scenes in area V6 (t(11) = 3.57, p<0.01). However, the level of this selectivity was significantly weaker in V6 compared to that in PIGS (t(11) = 2.63, p=0.02). Moreover, in the group-averaged activity maps, the contrast between coherently vs. incoherently changing scenes yielded a stronger response outside (rather than inside) the POS and also in area MT, located at the tip of medial

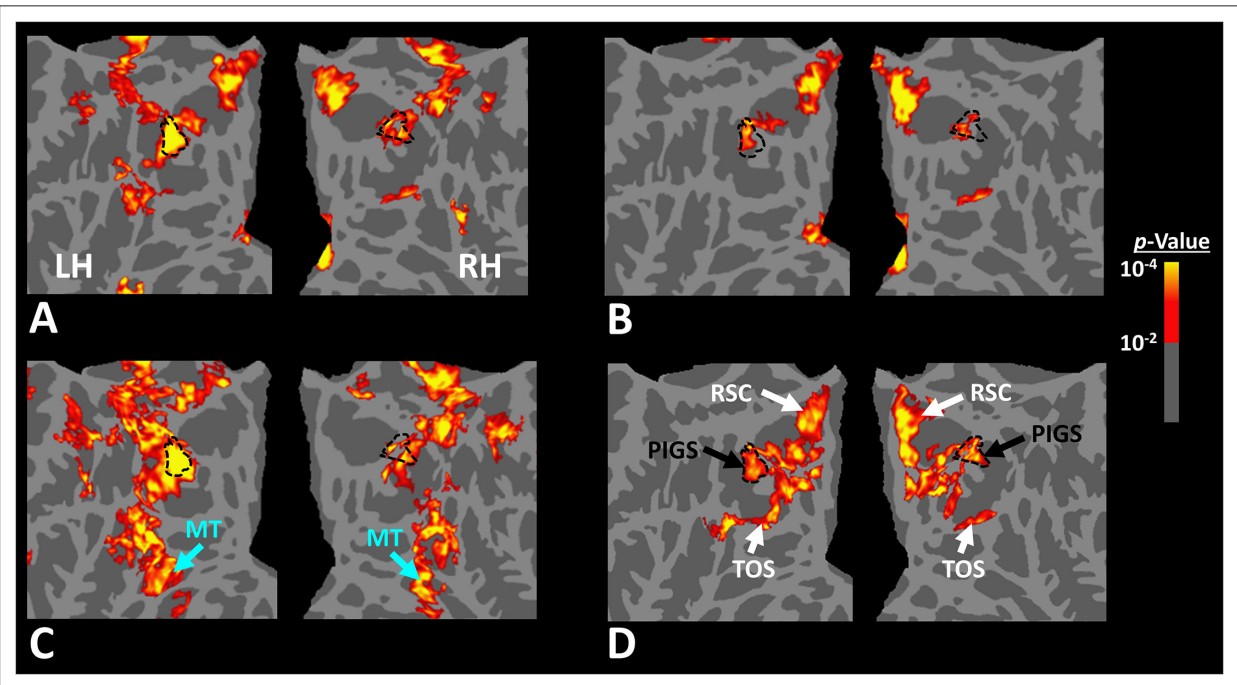

**Figure 11.** Scene-selective response to coherently vs. incoherently changing scenes within the intraparietal region (Experiment 6). Panels (**A**) and (**B**) show respectively the group-averaged activity evoked by coherently and incoherently changing scenes relative to faces. Panel (**C**) shows the group-averaged response evoked by the 'coherently > incoherently changing scenes' contrast. Among scene-selective areas, only posterior intraparietal gyrus scene-selective site (PIGS) showed significant sensitivity to the observer ego-motion. Besides PIGS, this contrast also evoked activity within area MT (cyan arrows), also within more dorsal portions of the parietal cortex. Panel (**D**) shows the location of scene-selective areas in the same group of subjects based on an independent set of scene and face stimuli (Experiment 1). In all panels, the location of PIGS outside the parieto-occipital sulci (POS) defined based on panel (**D**) is indicated by black dashed lines. All maps were generated based on random-effects, after correction for multiple comparisons.

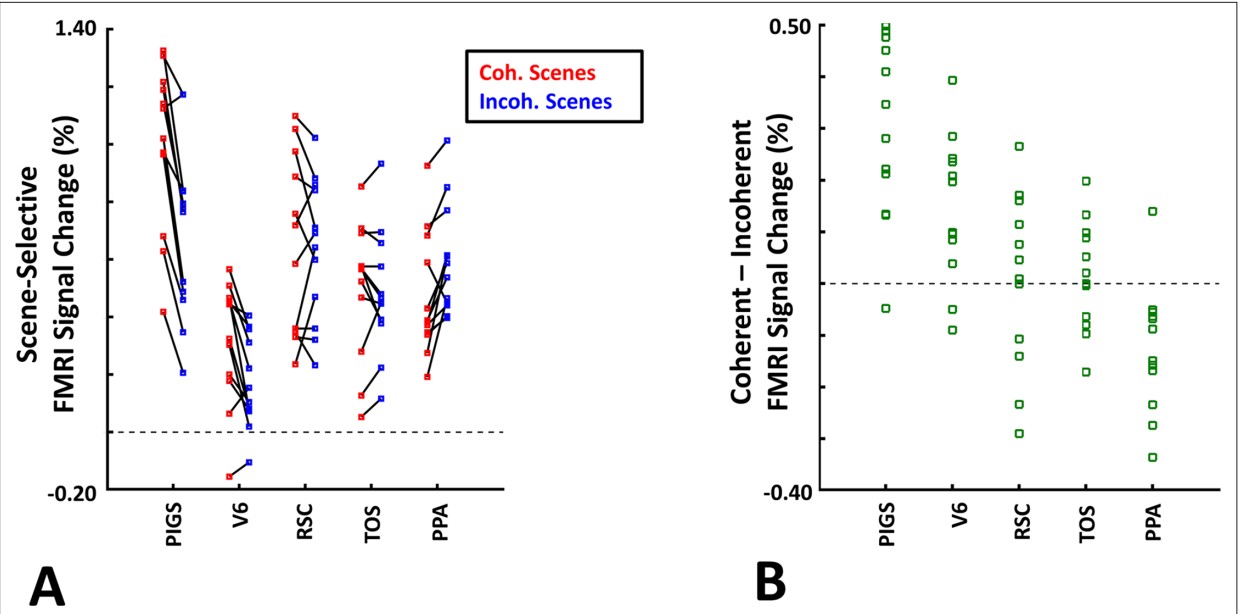

**Figure 12.** The scene-selective activity evoked within posterior intraparietal gyrus scene-selective site (PIGS) is influenced by the ego-motion. Panel (**A**) shows the scene-selective activity evoked by the coherently (red) and incoherently changing scenes (blue), measured relative to the response to the faces, across areas PIGS, V6, restrosplenial cortex/medial place area (RSC/MPA), transverse occipital sulcus/occipital place area (TOS/OPA), and parahippocampal place area/temporal place area (PPA/TPA). Panel (**B**) shows the level of difference between the response evoked by 'coherently – incoherently changing scenes' across the regions of interest. While all regions showed a significantly stronger response to scenes compared to faces, PIGS showed the strongest impact of ego-motion on the scene-selective response. Other details are similar to *Figure 7*.

temporal sulcus (*Figure 11C*). Together, these results suggest that the impact of ego-motion on scene processing is stronger in PIGS than that in V6.

In the same session (but different runs), we also tested the selectivity of the PIGS response for simpler forms of motion. In different blocks, subjects were presented with radially moving vs. stationary concentric rings (see 'Methods'). Consistent with the previous studies of motion perception (*Pitzalis et al., 2010*; *Korkmaz Hacialihafiz and Bartels, 2015*), the results of an ROI analysis here did not yield any strong (significant) motion-selective activity within PIGS (t(11) = 1.84, p=0.10), RSC/MPA (t(11) = 1.97, p=0.08), PPA/TPA (t(11) = 1.93, p=0.08), and V6 (t(11) = 2.03, p=0.07). In contrast, we found strong motion selectivity within area TOS/OPA (t(11) = 4.57, p<10$^{-3}$), likely due to its overlap with the motion-selective area V3A/B (*Nasr et al., 2011*). Thus, in contrast to optic flow and ego-motion, simpler forms of motion only evoke weak-to-no selective activity within PIGS and V6.

## Experiment 7: PIGS response to biological motion

The results of Experiment 6 showed that PIGS responds selectively to ego-motion in scenes, but not strongly to radially moving rings. However, it could be argued that PIGS may also respond to the other types of complex motion, for example, biological motion. To test this hypothesis, we measured the PIGS response to biological vs. translational motion in 12 subjects (see 'Methods'). As illustrated in *Figure 13*, and consistent with the previous studies of biological motion (*Puce et al., 1998*; *Beauchamp et al., 2003*; *Puce and Perrett, 2003*; *Pelphrey et al., 2005*; *Jastorff and Orban, 2009*; *Kamps et al., 2016*), biological motion evoked a stronger response bilaterally within area MT and superior temporal sulcus but not within the posterior intraparietal gyrus. Consistent with the maps, an ROI analysis (based on the functionally defined labels) showed no significant difference between the response to biological vs. translational motion within PIGS (t(11) = 1.27, p=0.23), TOS/OPA (t(11) = 1.63, p=0.13), RSC/MPA (t(11) = 1.40, p=0.18), and PPA/TPA (t(11) = 0.41, p=0.69). These results indicated that PIGS does not respond to all types of complex motion.

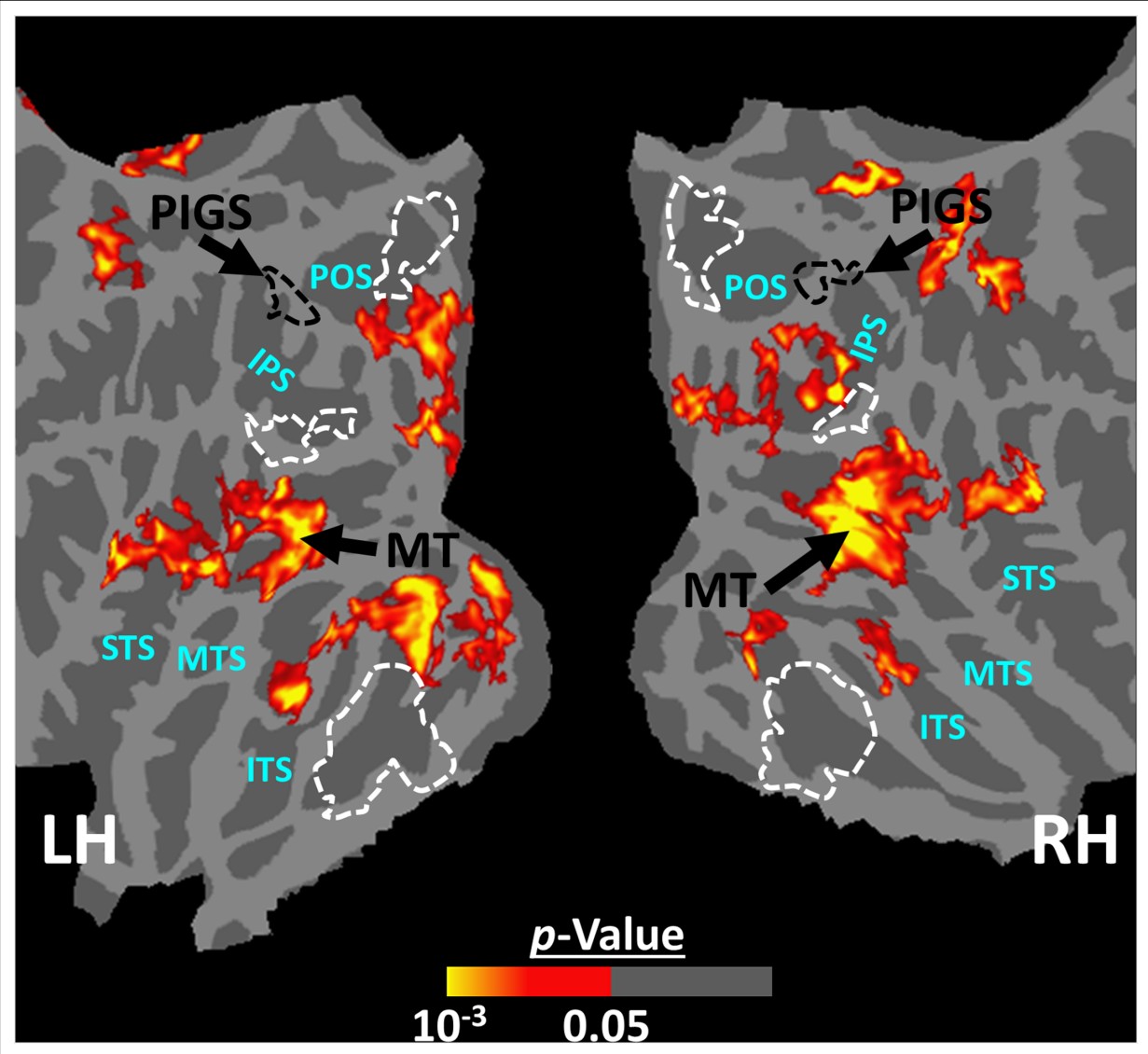

**Figure 13.** The group-averaged activity map evoked by the 'biological > translational motion' contrast. Despite the low threshold used to generate these maps, we did not detect any significant activity evoked by the 'biological > translational motion' contrast within the posterior intraparietal gyrus scene-selective site (PIGS) and/or the other scene-selective areas. Rather, this contrast evoked a significant activity mainly within the inferior temporal sulcus (ITS), medial temporal sulcus (MTS), and superior temporal sulcus (STS). In both hemispheres, the location of PIGS, detected based on an independent set of scene and face stimuli (as in Experiment 1), is indicated by black dashed lines. The location of other scene-selective areas is indicated by white dashed lines.

## Discussion

These data suggest that selective scene processing is not limited to areas PPA/TPA, RSC/MPA, and TOS/OPA, and that additional smaller scene-selective sites can also be found across the visual system. By focusing on one small scene-selective site, we showed that this site (PIGS) was consistently identifiable across individuals and groups. We also showed that inclusion of this site in the models of scene processing may clarify how ego-motion influences scene perception.

### FMRI and all that 'noise, noise, noise'!

The early fMRI studies dealt with a considerable amount of noise in measurements, partly due to using lower magnetic field scanners and imperfect hardware and software. This noise in measurements affected the reliability of the findings. Consequently, those early studies focused on larger activity

sites that were more reliably detectable across subjects/sessions. The smaller sites were either ignored or eliminated by excessive signal smoothing, applied to enhance the level of contrast-to-noise ratio.

However, advances in neuroimaging techniques have now made it possible to detect and distinguish fMRI activity at the spatial scale of cortical columns (*Yacoub et al., 2007*; *Zimmermann et al., 2011*; *Nasr et al., 2016a*). Although the reliability of the fMRI signal still depends on the number of trial repetitions, a spatially confined, but extensively repeated, evoked response can be detected reliably across different sessions (*Nasr et al., 2016a*; *Kennedy et al., 2023*).

The present data shows that PIGS could be localized consistently across multiple subjects and across different sessions and scanners. Furthermore, our results indicated that the probabilistic labels, generated based on one population, can be used to localize PIGS and distinguish its function from the adjacent regions (e.g., V6) in a second population. Together, these results highlight the reliability of current fMRI techniques in detecting smaller cortical regions in the level of individual subjects.

## PIGS responds selectively to a variety of scene stimuli

To establish a true category-selective response, the stimulus set should sample enough variety to reflect the range and variability among the category members. Consistent with this are the many (and continuing) studies seeking to define the range and fundamental aspects of 'place selective' (*Epstein and Kanwisher, 1998*; *Troiani et al., 2014*) and 'face selective' (*Kanwisher et al., 1997*; *Yue et al., 2011*) stimuli in extrastriate visual cortex, decades after their first discovery.

Accordingly, here we tested five different scene stimulus sets across our experiments, including a wide variety of indoor/outdoor and natural/manmade scenes. In all cases, we were able to evoke a selective response within PIGS, and the level of this response was comparable to that in the adjacent scene-selective areas RSC/MPA and TOS/OPA. Thus, the scene-selective response in PIGS appeared not to be limited to a single subset of scenes. However, it remains unclear whether scene stimuli are differentiable from each other based on the pattern of evoked response in this region. More experiments are necessary to test this hypothesis (see also the Limitations).

## PIGS and TOS/OPA are two different areas

Our results clearly showed that PIGS and TOS/OPA are two distinct scene-selective areas based on multiple criteria: first, anatomically, TOS/OPA is located mostly anterior to the IPS, whereas PIGS is located more dorsally and posterior to the IPS. Second, TOS/OPA overlaps with areas V3A/B and IPS0, whereas PIGS was located adjacent to IPS3-4. Third, these two areas respond distinctly to moving stimuli. Specifically, while TOS/OPA responds selectively to moving concentric rings and less selectively to ego-motion, PIGS shows the opposite pattern and responds selectively to ego-motion within the naturalistic scenes but not to moving rings (see below). Considering these anatomical and functional differences, these two areas appear to be two distinct hubs within the scene processing networks.

Also notably, PIGS is located relatively far from the lateral place memory area (LPMA), which is located anterior to the IPS and close to the tip of the superior temporal sulcus (*Steel et al., 2021*; *Steel et al., 2023*). Considering this, and the fact that there was no memory demand in our paradigms, PIGS and LPMA also appear to be two distinct visual areas.

## PIGS is not just another scene-selective area

Our results (Experiment 6) suggest that ego-motion can significantly influence the activity evoked within PIGS. This phenomenon distinguishes the role of PIGS in scene perception, relative to other scene-selective regions. Specifically, previous studies have shown that PPA/TPA and RSC/MPA show weak-to-no sensitivity to motion per se (*Korkmaz Hacialihafiz and Bartels, 2015*). In comparison, area TOS/OPA shows a stronger motion-selective response, presumably related to its (partial) overlap with area V3A/B (*Tootell et al., 1997*; *Nasr et al., 2011*). Instead, the current data show that the ego-motion-related activity within PIGS is stronger than in TOS/OPA.

This finding is consistent with the fact that PIGS is located adjacent to area V6 (*Figures 4 and 5*), an area that contributes to encoding optic flow (*Pitzalis et al., 2010*). Considering PIGS and V6 proximity, hypothetical inputs from V6 may contribute to the strong ego-motion selective response in PIGS. This said, the current data also suggests that the role of PIGS differs from that in V6 in terms of ego-motion encoding. Compared to V6, PIGS showed a stronger impact of ego-motion on

scene processing, while V6 shows a stronger response to optic flow induced by random dot arrays. Thus, PIGS contributes to scene encoding and ego-motion within scenes, while V6 is likely involved in detecting optic flow caused by ego-motion.

## Ego-motion encoding in PIGS vs. TOS/OPA

We showed that PIGS and TOS/OPA are located on two different sides of the IPS with TOS/OPA located more ventrally compared to PIGS. We also showed a stronger impact of ego-motion on activity within PIGS compared to TOS/OPA. In contrast, TOS/OPA (but not PIGS) responded selectively to simpler forms of motion. These results suggest that PIGS and TOS/OPA are likely two different visual areas, with PIGS being involved in encoding higher-level ego-motion cues.

However, at least two previous studies suggested that area TOS/OPA may also contribute to ego-motion encoding in scenes. Specifically, Kamps and colleagues have shown increased response in TOS/OPA during ego-motion vs. static scene presentation (*Kamps et al., 2016*). *Jones et al., 2023* have also shown that ego-motion (and not other types of movements) enhances TOS/OPA activity when compared to scrambled scenes. In contrast to these findings, our tests showed weak-to-no ego-motion-related activity enhancement in area TOS/OPA.

This difference may well reflect methodological discrepancies. Specifically, in the study by Kamps et al., the static and ego-motion stimuli were presented with two different refresh rates. While in our study, the coherently and incoherently changing stimuli were refreshed with the same temporal frequency (see 'Methods'). In the study by *Jones et al., 2023*, the response to scrambled scenes was used as a control condition, whereas our stimuli were more equivalent, differing only in the sequence of image presentation. Moreover, these studies used higher levels of spatial smoothing (FWHM = 5 mm) compared to the values we used here during preprocessing. Also, for understandable reasons, they limited their analysis to previously known scene-selective areas. These technical differences make it difficult to directly compare the two sets of results.

## Ego-motion but not attention and/or visual context

Experiment 6 showed stronger scene-selective activity within PIGS when subjects were presented with coherently (compared to incoherently) changing scenes. It could be argued that coherently changing scenes attract more attention compared to incoherently changing scenes. On the face of it, this hypothesis appears to be consistent with the expected contribution of the intraparietal cortex in controlling spatial attention (*Behrmann et al., 2004*; *Szczepanski et al., 2010*). But if true, attention to scenes should also increase the level of activity within the scene-selective areas (*O'Craven et al., 1999*; *Nasr and Tootell, 2012b*; *Baldauf and Desimone, 2014*). While here, we did not find any significant activity increases in response to coherently (vs. incoherently) changing scenes in PPA/TPA, RSC/MPA, and TOS/OPA. Thus, modulation of attention, per se, could not be responsible for the enhanced activity within PIGS in response to coherently (compared to incoherently) changing scenes.

Furthermore, the stimuli used in coherently vs. incoherently changing block conditions represented the same scenes, and the only difference between the two conditions was the sequence of images within the blocks. In that sense, the two experimental conditions may be considered to have the same visuospatial context. However, it could be also argued that the coherently changing scenes provide more information about the environmental layout. In that case, considering the previous reports that PPA/TPA and RSC/MPA are also involved in layout encoding (*Epstein and Kanwisher, 1998*; *Wolbers et al., 2011*), we expected to see more activity within those regions in response to coherently compared incoherently changing scenes. In the absence of such an activity modulation, a change in the visual context could not be responsible for the enhanced PIGS activity.

## Direction-selective response within the intraparietal cortex

Motion-selective sites are expected to show at least some level of sensitivity to motion direction (*Albright et al., 1984*; *Zimmermann et al., 2011*). We did not test the sensitivity of PIGS to the direction of ego-motion. However, *Pitzalis et al., 2020* have shown evidence for motion direction encoding within the V6+ region (*Pitzalis et al., 2020*). Furthermore, Tootell et al. reported evidence for motion direction (approaching vs. withdrawing) encoding within posterior intraparietal cortex (*Tootell et al., 2022*). Although none of these studies showed any evidence for a new scene-selective area, they

raised the possibility that PIGS may also contribute toward encoding ego-motion direction, and even higher-level cognitive concepts such as detecting an intrusion to personal space (*Holt et al., 2014*).

### Limitations

In the past, many studies have scrutinized the response function of scene-selective areas to numerous stimulus contrasts. According to these studies, scene-selective areas can differentiate many object categories based on their low-, mid-, and/or higher-level visual features such as their natural size (*Konkle and Oliva, 2012*), (non-)animacy (*Yue et al., 2020*; *Coggan and Tong, 2023*), rectilinearity (*Nasr et al., 2014*), spatial layout (*Harel et al., 2013*), orientation (*Nasr and Tootell, 2012a*), spikiness (*Coggan and Tong, 2023*), location within the visual field (*Levy et al., 2001*), and spatial content (*Bar et al., 2008*). Our findings are only a *first* step toward characterizing PIGS in greater detail. More tests are required to reach the current (yet incomplete) knowledge about the response function of PIGS.

### Conclusion

Neuroimaging studies of scene perception have typically focused on linking scene perception to the evoked activity within PPA/TPA, TOS/OPA, and RSC/MPA. Although other scene-selective sites are detectable across the visual cortex, they are largely ignored because of their relatively small size. Our data suggests that the future inclusion of these small sites in the models of scene perception may help clarify current models of scene processing in dynamic environments.

## Methods

### Participants

Fifty-nine human subjects (33 females), aged 22–68 y, participated in this study. All subjects had normal or corrected-to-normal vision and radiologically normal brains, without any history of neuropsychological disorder. All experimental procedures conformed to NIH guidelines and were approved by the institutional review board of the Massachusetts General Hospital (2018P001557). Written informed consent was obtained from all subjects before the experiments.

### General procedure

This study consists of seven experiments during which we used fMRI to localize and study the evoked scene-selective responses. During these experiments, stimuli were presented via a projector (1024 × 768 pixel resolution, 60 Hz refresh rate) onto a rear-projection screen. Subjects viewed the stimuli through a mirror mounted on the receive coil array. Details of these stimuli are described in the following sections.

During all experiments, to ensure that subjects were attending to the screen, they were instructed to report color changes (red to blue and vice versa) for a centrally presented fixation object (0.1° × 0.1°) by pressing a key on the keypad. Subject detection accuracy remained above 75% and showed no significant difference across experimental conditions (p>0.10). MATLAB (MathWorks, Natick, MA) and the Psychophysics Toolbox (*Brainard, 1997*; *Pelli, 1997*) were used to control stimulus presentation.

#### Experiment 1: Localization of scene-selective areas

In 14 subjects (six females), we localized scene-selective areas PPA/TPA, RSC/MPA, and TOS/OPA by measuring their evoked brain activity using a 3T fMRI scanner as they were presented with eight colorful images of real-world (indoor) scenes vs. (group) faces (*Nasr et al., 2011*). Scene and face images were retinotopically centered and subtended 20° × 26° of visual field without any significant differences between their root mean square (RMS) contrast (t(14) = 1.10, p=0.29). Scene and face stimuli were presented in different blocks (16 s per block and 1 s per image). Each subject participated in four runs and each run consisted of 10 blocks plus 32 s of blank presentation at the beginning and at the end of each block. Within each run, the sequence of blocks and the sequence of images within them was randomized.

#### Experiment 2: Reproducibility of PIGS across scan sessions (3T vs. 7T)

To localize PIGS with higher spatial resolution and enhance the signal-/contrast-to-noise ratio (relative to Experiment 1), four subjects were randomly selected from those who participated in Experiment

1 and were scanned using a 7T scanner. These individuals were presented with 300 grayscale images of scenes and 48 grayscale images of (single) faces other than those used in Experiment 1. Here, scene images included pictures of indoor (100 images), manmade outdoor (100 images) and natural outdoor (100 images) scenes, selected from the Southampton-York Natural Scenes dataset (*Adams et al., 2016*).

As in Experiment 1, all images were retinotopically centered and subtended 20° × 26° of visual field, and there was no significant difference between the RMS contrast across the two categories (t(346) = 0.75, p=0.38). Scene and face images were presented across different blocks. Each block contained 24 stimuli (1 s per stimuli), with no blank presentation between the stimuli. The sequence of stimuli was randomized within the blocks. Each subject participated in 12 runs (11 blocks per run; 24 s per block; 1 s per stimulus), beginning and ending with an additional block (12 s) of uniform black presentation. In each run, the sequence of blocks and the sequence of images within them were randomized.

## Experiment 3: PIGS localization relative to area V6 and retinotopic visual areas

Experiment 3a was designed to clarify the relative localization of PIGS vs. area V6 (*Pitzalis et al., 2010*). All 14 subjects who participated in Experiment 1 were examined again in a separate scan session using a 3T scanner. During this scan session, we localized area V6 by contrasting the response evoked by coherent radially moving (optic flow) vs. randomly moving white dots (20° × 26°), presented against a black background. The experiment was block-designed, and each block took 16 s. Each subject participated in five runs (14 blocks per run), beginning and ending with an additional block of 16 s uniform black presentation.

Experiment 3b was designed to compare the localization of PIGS relative to the border of retinotopic visual areas such as V3A/B and IPS0-4. Two subjects who had participated in Experiment 2 were randomly selected and scanned again in a 7T scanner, during which we defined the border of retinotopic visual areas using a phase encoding approach (*Sereno et al., 1995*; *Engel et al., 1997*). Specifically, subjects were presented with rotating (CW and CCW) wedge-shaped (45°) apertures that revolved over 28 s, followed by a 4 s blank presentation. Instead of using a flashing checkerboard, we used naturalistic stimuli consisting of color objects presented against a pink-noise background, updated at 15 Hz (*Benson et al., 2018*). Each subject participated in 10 runs (four blocks per run).

## Experiment 4: Localization of PIGS in a larger population

Considering the small size of PIGS, it was important to show that this area could survive group-averaging over larger populations, compared to Experiment 1. Accordingly, Experiment 4 localized this area in a large pool of subjects, consisting of 31 individuals (19 females) other than those who participated in Experiment 1. The stimuli and procedure were identical to Experiment 1.

## Experiment 5: Response to two independent sets of scenes and non-scene objects

Experiments 1–4 used the response evoked by scenes vs. faces to localize PIGS. However, it remained unknown whether PIGS also showed a selective response to the 'scenes vs. objects' contrast. Accordingly, in two independent groups of subjects (no overlap), Experiment 5 tested the response evoked by scenes vs. non-scene objects in PIGS and the adjacent areas (i.e., V6, TOS/OPA and RSC/MPA).

Specifically, in Experiment 5a, 13 subjects (seven females), other than those who participated in Experiment 1, were scanned using a 3T scanner. They were presented with 22 grayscale images of indoor/outdoor scenes, other than those presented in Experiments 1–4, and 88 grayscale images that included either a single or multiple everyday non-animate (non-face) objects (*Nasr and Tootell, 2012a*; *Nasr et al., 2014*). All stimuli were retinotopically centered and presented within a circular aperture (diameter = 20°). The RMS contrast of the objects was significantly higher than the scenes (t(108) = 3.72, p<10⁻³). Scene and object images were presented in different blocks according to their category (22 s per block and 1 s per image). Each subject participated in 12 runs and each run consisted of 9 blocks, plus 16 s of blank presentation at the beginning and the end of each block. As in other experiments, the sequence of blocks and the sequence of images within them were randomized.

In Experiment 5b, 14 subjects (eight females), other than those who participated in Experiments 1 and 5a, were scanned using a 3T scanner. Each subject was presented with 32 grayscales images of indoor/outdoor scenes, 32 images of everyday (non-face) objects plus also their scrambled versions, and 32 images of single faces (*Nasr and Rosas, 2016b*). Scene and non-scene stimuli were different than those used in Experiments 1–4 and 5a. In contrast to Experiment 5a, all non-scene images included only one single object and there was no significant difference between the RMS contrasts of scenes and the three object categories (F(3, 111) = 0.42, p=0.74). Other details were similar to those in Experiment 5a.

## Experiment 6: Coherently vs. incoherently changing scenes

This experiment was designed to differentiate the role of PIGS in scene perception from TOS/OPA, RSC/MPA, and PPA/TPA. In total, 12 subjects, from the 14 subjects who participated in Experiment 1, participated in this experiment. The excluded two subjects could not participate further in our tests for personal reasons. Subjects were scanned using a 3T scanner on a different day relative to Experiments 1–3. During this scan, they were presented with rapidly 'coherently vs. incoherently changing scenes' (100 ms per image), across different blocks (16 s per block).

Coherently changing scenes implied ego-motion (fast walking) along three different outdoor natural trails. Stimuli (20° × 26°) were generated as one of the experimenters walked through the trails while carrying a camera mounted on his forehead, taking pictures every 2 m. Incoherently changing scenes consisted of the same images as the coherently changing blocks, but with randomized order. In other words, the only difference between the coherently vs. incoherently changing scenes was the sequence of stimuli within the block. For both coherently and incoherently changing scenes, images from different trails were presented across different blocks.

In separate blocks, subjects were also presented with 80 images that included multiple faces (20° × 26°) with the same timing as the scene images (i.e., 100 ms per image; 16 s per block). All stimuli were grayscaled. Each subject participated in six runs and each run consisted of nine blocks, plus 8 s of blank presentation at the beginning and the end of each block and 4 s of blank presentation between blocks.

On different runs (within the same session), subjects were also presented with concentric rings, extending 20° × 26° (height × width) in the visual field, presented against a light gray background (40 cd/m$^2$). In half of the blocks (16 s per block), rings moved radially (centrifugally vs. centripetally; 4°/s) and the direction of motion changed every 4 s to reduce the impact of motion aftereffects. In the remaining half of the blocks, rings remained stationary throughout the whole block. Each subject participated in two runs and each run consisted of eight blocks, plus 16 s of uniform gray presentation at the beginning and the end of each run. The sequence of moving and stationary blocks was pseudo-randomized across runs.

## Experiment 7: Response to biological motion

To test whether PIGS also responds selectively to biological motion, 12 individuals were selected randomly and scanned using a 3T scanner while they were presented with the moving point-lights that represented complex biological movements such as crawling, cycling, jumping, paddling, and walking (*Jastorff and Orban, 2009*). Each action was presented for 2 s and the sequence of actions was randomized across the blocks (20 s per block). As a control, in different blocks, the subjects were shown the same stimuli when all of the point-lights moved in the same direction (i.e., translation motion). Each subject participated in 11 runs and each run consisted of 12 blocks, plus 10 s of blank presentation at the beginning and the end of each run.

## Imaging

### 3T scans

In Experiments 1, 3a, and 4–6, subjects were scanned using a horizontal 3T scanner (Tim Trio, Siemens Healthcare, Erlangen, Germany). Gradient echo EPI sequences were used for functional imaging. Functional data were acquired using single-shot gradient echo EPI with nominally 3.0 mm isotropic voxels (TR = 2000 ms; TE = 30 ms; flip angle = 90°; band width [BW] = 2298 Hz/pix; echo-spacing = 0.5 ms; no partial Fourier; 33 axial slices covering the entire brain; and no acceleration). During the first 3T scan (see 'Methods'), structural (anatomical) data were acquired for each subject using a 3D

T1-weighted MPRAGE sequence (TR = 2530 ms; TE = 3.39 ms; TI = 1100 ms; flip angle = 7°; BW = 200 Hz/pix; echo-spacing = 8.2 ms; voxel size = 1.0 × 1.0 × 1.33 mm).

## 7T scans

In Experiments 2 and 3b, subjects were scanned using a 7T Siemens whole-body scanner (Siemens Healthcare) equipped with SC72 body gradients (maximum gradient strength, 70 mT/m; maximum slew rate, 200 T/m/s) using a custom-built 32-channel helmet receive coil array and a birdcage volume transmit coil. Voxel dimensions were nominally 1.0 mm, isotropic. Single-shot gradient-echo EPI was used to acquire functional images with the following protocol parameter values: TR = 3000 ms; TE = 28 ms; flip angle = 78°; BW = 1184 Hz/pix; echo-spacing = 1 ms; 7/8 phase partial Fourier; 44 oblique-coronal slices; and acceleration factor *r* = 4 with GRAPPA reconstruction and FLEET-ACS data (*Polimeni et al., 2016*) with 10° flip angle. The field of view included the occipital-parietal brain areas to cover PIGS, RSC/MPA, and TOS/OPA (but not PPA/TPA).

## Data analysis

### Structural data analysis

For each subject, inflated and flattened cortical surfaces were reconstructed based on the high-resolution anatomical data (*Dale et al., 1999*; *Fischl et al., 1999*; *Fischl et al., 2002*), during which the standard pial surface was generated as the gray matter border with the surrounding cerebrospinal fluid or CSF (i.e., the GM–CSF interface). The white matter surface was also generated as the interface between white and gray matter (i.e., WM–GM interface). In addition, an extra surface was generated at 50% of the depth of the local gray matter (*Dale et al., 1999*).

### Individual-level functional data analysis

All functional data were rigidly aligned (6 df) relative to subject's own structural scan using rigid Boundary-Based Registration (*Greve and Fischl, 2009*), and then motion-corrected. Data collected in the 3T (but not 7T) scanner was spatially smoothed using a 3D Gaussian kernel (2 mm FWHM). To preserve the spatial resolution, data collected within the 7T scanner was not spatially smoothed.

Subsequently, a standard hemodynamic model based on a gamma function was fit to the fMRI signal to estimate the amplitude of the BOLD response. For each individual subject, the average BOLD response maps were calculated for each condition (*Friston et al., 1999*). Finally, voxel-wise statistical tests were conducted by computing contrasts based on a univariate general linear model.

The resultant significance maps based on 3T scans were sampled from the middle of cortical gray matter defined for each subject based on their structural scan (see 'Methods'). For 7T scans, the resultant significance maps were sampled from deep cortical layers at the gray–white matter interface. This procedure reduced the spatial blurring caused by superficial veins (*Koopmans et al., 2010*; *Polimeni et al., 2010*; *De Martino et al., 2013*; *Nasr et al., 2016a*). For presentation, the resultant maps were projected either onto the subject's reconstructed cortical surfaces or onto a common template (fsaverage; FreeSurfer; *Fischl, 2012*).

### Group-level functional data analysis

To generate group-averaged maps, functional maps were spatially normalized across subjects, then averaged using weighted least square random-effects models (using the contrast effect size and the variance of contrast effect size as the input parameters) and corrected for multiple comparisons (*Friston et al., 1999*). For *Figure 1A* and to replicate our original finding (*Nasr et al., 2011*), the group-average maps were generated using fixed-effects. The resultant significance maps were projected onto a common human brain template (fsaverage).

### ROI analysis

The main ROIs included area PIGS, the two neighboring scene-selective areas (RSC/MPA, TOS/OPA), and area V6. In Experiment 6, we also included area PPA/TPA in our analysis. These ROIs were localized in two different ways: (1) functionally, for each subject based on their own evoked activity (see below), and (2) probabilistically, based on activity measured in a different group of subjects.

## Functionally localized ROIs

For those subjects who participated in Experiments 6 and 7, we localized scene-selective areas PIGS, TOS/OPA, RSC/MPA, and PPA/TPA based on their stronger response to scenes compared to faces at a threshold level of $p<10^{-2}$ using the method described in Experiment 1. For subjects in Experiment 6, we also localized area V6 based on the expected selective response in this region to coherent radially vs. incoherently moving random dots (see Experiment 3). In those subjects in which PIGS and V6 showed partial overlap, the overlapping parts were excluded for the analysis.

## Probabilistically localized ROIs

For those subjects who participated in Experiments 4 and 5, we tested the consistency of PIGS locations across populations using probabilistic labels for areas PIGS, TOS/OPA, RSC/MPA, and V6. These labels were generated based on the results of Experiment 1 (for PIGS, TOS/OPA, and RSC/MPA) and Experiment 3a (for V6). Specifically, we localized the ROIs separately for the individual subjects who participated in Experiments 1 and 3a. Then the labels were overlaid on a common brain template (fsaverage). We computed the probability that each vortex within the cortical surface belonged to one of the ROIs. The labels for PIGS, TOS/OPA, RSC/MPA, and V6 were generated based on those vertices that showed a probability higher than 20%. This method assured us that our measurements were not biased by those subjects who showed stronger scene-selective responses. Moreover, by selecting a relatively low threshold (i.e., 20%), we avoided confining our ROIs to the center of activity sites.

## Statistical tests

To test the effect of independent parameters, we applied paired *t*-tests and/or a repeated-measures ANOVA, with Greenhouse–Geisser correction whenever the sphericity assumption was violated.

# Acknowledgements

This work was supported by NIH NEI (grants R01 EY017081 and R01 EY030434) and the MGH/HST Athinoula A Martinos Center for Biomedical Imaging. Crucial resources were made available by a NIH Shared Instrumentation Grant S10-RR019371. We thank Ms. Azma Mareyam for help with hardware maintenance during this study. We also thank Dr. Claudio Galletti for his helpful comments.

# Additional information

## Funding

| Funder | Grant reference number | Author |
| --- | --- | --- |
| National Institutes of Health | R01EY017081 | Roger BH Tootell |
| National Institutes of Health | R01EY030434 | Shahin Nasr |

The funders had no role in study design, data collection and interpretation, or the decision to submit the work for publication.

## Author contributions

Bryan Kennedy, Formal analysis, Visualization, Writing – original draft; Sarala N Malladi, Visualization, Writing – review and editing; Roger BH Tootell, Writing – original draft, Writing – review and editing; Shahin Nasr, Conceptualization, Resources, Data curation, Software, Formal analysis, Supervision, Funding acquisition, Validation, Investigation, Visualization, Methodology, Writing – original draft, Project administration, Writing – review and editing

## Author ORCIDs

Shahin Nasr https://orcid.org/0000-0002-7546-9976

### Ethics

Human subjects: All experimental procedures conformed to NIH guidelines and were approved by Massachusetts General Hospital protocols. Written informed consent was obtained from all subjects before the experiments. (2018P001557).

Reviewer #2 (Public Review): https://doi.org/10.7554/eLife.91601.3.sa1
Reviewer #3 (Public Review): https://doi.org/10.7554/eLife.91601.3.sa2
Author Response https://doi.org/10.7554/eLife.91601.3.sa3

---

## Additional files

### Supplementary files

• MDAR checklist

### Data availability

The stimuli, significance maps, generated probabilistic labels, and the measured activity across ROIs, as presented in Figures 1-13, are already available to the public (https://doi.org/10.5061/dryad. xwdbrv1mj; see also https://mesovision.martinos.org/datasets). MATLAB (RRID:SCR_001622; https:// www.mathworks.com); FreeSurfer (RRID:SCR_001847; https://surfer.nmr.mgh.harvard.edu/fswiki/ FsFast); and Psychophysics Toolbox (RRID:SCR_002881; http://psychtoolbox.org/docs/Psychtoolbox).

The following dataset was generated:

| Author(s) | Year | Dataset title | Dataset URL | Database and Identifier |
|---|---|---|---|---|
| Nasr S | 2024 | A previously undescribed scene-selective site is the key to encoding ego-motion in naturalistic environments | https://doi.org/ 10.5061/dryad. xwdbrv1mj | Dryad Digital Repository, 10.5061/dryad.xwdbrv1mj |

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
