## [Editor Report · eLife assessment]

In this article, the authors present a wealth of fMRI data at both 3T and 7T to identify a scene-selective region of the intraparietal gyrus (‘PIGS’) that appears to have some responsivity to characteristics of ego-motion. In a series of experiments, they delineate the anatomical location of PIGS and functionally differentiate it from nearby V6 and OPA. Evidence for these **important** findings is **solid**, but further investigations as to the role of this region in processing ego-motion will be needed to confirm this conclusion.

---

## [Referee Report · Reviewer #2 (Public Review)]

Summary

The authors report an extensive series of neuroimaging experiments (at both 3T and 7T) to provide evidence for a scene-selective visual area in human posterior parietal cortex (PIGS) that is distinct from the main three (parahippocampal place area, PPA; occipital place area, OPA; medial place area, MPA) typically reported in the literature. Further, they argue that in comparison with the other three, this region may specifically be involved in representing ego-motion in natural contexts. The characterization of this scene-selective region provides a useful reference point for studies of scene processing in humans.

Strengths

One of the major strengths of the work is the extensive series of experiments reported, showing clear reproducibility of the main finding and providing functional insight into the region studied. The results are clearly presented and convincing with careful comparison to retinotopic and scene-selective regions described in prior studies.

Weaknesses

While the results are strong and clear, the claim in the title ("A previously undescribed scene-selective site is the key to encoding ego-motion in naturalistic environments") is not fully supported. The results show that this scene-selective region is sensitive to visual cues that reflect ego-motion but not that it is "key" to encoding ego-motion. Further, there are many differences between the two types of stimuli used to test ego-motion and greater characterization of this scene-selective region will be needed to confirm this conclusion.

---

## [Referee Report · Reviewer #3 (Public Review)]

Summary:

The authors report a scene-selective areas in the posterior intraparietal gyrus (PIGS). This area lies outside the classical three scene-selective regions (PPA/TPA, RSC/MPA, TOS/OPA), and is selective for ego motion.

Strengths:

The authors firmly establish the location and selectivity of the new area through a series of well-crafted controlled experiments. They show that the area can be missed with too much smoothing, thus providing a case for why it has not been previously described. They show that it appears in much the same location in different subjects, with different magnetic field strengths, and with different stimulus sets. Finally, they show that it is selective for ego motion - defined as series of sequential photographs of an egocentric trajectory along a path. They further clarify that the area is not generically motion selective by showing that it does not respond to biological motion without an egomotion component to it. All statistics are standard and sound; the evidence presented is strong.

Weaknesses:

There are a few weaknesses in this work. If pressed, I might say that the stimuli depicting ego motion do not, strictly speaking, depict motion, but only apparent motion between 2s apart photographs. However, this choice was made to equate frame rates and motion contrast between the 'ego motion' and a control condition, which is a useful and valid approach to the problem.

This is a very strong paper.

---

## [Author Response]

The following is the authors’ response to the original reviews.

**Reviewer 1:**
Comment 1.1: The distinction of PIGS from nearby OPA, which has also been implied in navigation and ego-motion, is not as clear as it could be.

Response1.1: The main “functional” distinction between TOS/OPA and PIGS is that TOS/OPA responds preferentially to moving vs. stationary stimuli (even concentric rings), likely due to its overlap with the retinotopic motion-selective visual area V3A, for which this is a defining functional property (e.g. Tootell et al., 1997, J Neurosci). In comparison, PIGS does not show such a motion-selectivity. Instead, PIGS responds preferentially to more complex forms of motion within scenes.

Moreover, PIGS and TOS/OPA are located in differently relative to the retinotopic visual areas. Briefly, PIGS is located adjacent to areas IPS3-4 while TOS/OPA overlaps with areas V3A/B and IPS0 (V7). This point is now highlighted in the new experiment 3b and the new Figure 6.In this revision, we also tried to better highlight these point in sections 4.3, 4.4 and 4.5. (see also the response to the first comment from Reviewer #2).

**Reviewer 2:**
Comment 2.1: First, the scene-selective region identified appears to overlap with regions that have previously been identified in terms of their retinotopic properties. In particular, it is unclear whether this region overlaps with V7/IPS0 and/or IPS1. This is particularly important since prior work has shown that OPA often overlaps with v7/IPS0 (Silson et al, 2016, Journal of Vision). The findings would be much stronger if the authors could show how the location of PIGS relates to retinotopic areas (other than V6, which they do currently consider). I wonder if the authors have retinotopic mapping data for any of the participants included in this study. If not, the authors could always show atlas-based definitions of these areas (e.g. Wang et al, 2015, Cerebral Cortex).

Response 2.1: We thank the reviewers for reminding us to more clearly delineate this issue of possible overlap, including the information provided by Silson et al, 2016. The issue of possible overlap between area TOS/OPA and the retinotopic visual areas, both in humans and non-human primates, was also clarified by our team in 2011 (Nasr et al., 2011). As you can see in Figure 6 (newly generated), and consistent with those previous studies, TOS/OPA overlaps with visual areas V3A/B and V7. Whereas PIGS is located more dorsally close to IPS3-4. As shown here, there is no overlap between PIGS and TOS/OPA and there is no overlap between PIGS and areas V3A/B and V7.

To more directly address the reviewer’s concern, in this revision, we have added a new experiment (Experiment 3b) in which we have shown the relative position of PIGS and the retinotopic areas in two individual subjects (Figure 6). All the relevant points are also discussed in section 4.3.

Comment 2.2: Second, recent studies have reported a region anterior to OPA that seems to be involved in scene memory (Steel et al, 2021, Nature Communications; Steel et al, 2023, The Journal of Neuroscience; Steel et al, 2023, biorXiv). Is this region distinct from PIGS? Based on the figures in those papers, the scene memory-related region is inferior to V7/IPS0, so characterizing the location of PIGS to V7/IPS0 as suggested above would be very helpful here as well. If PIGS overlaps with either of V7/IPS0 or the scene memory-related area described by Steel and colleagues, then arguably it is not a newly defined region (although the characterization provided here still provides new information).

Response 2.2: The lateral-place memory area (LPMA) is located on the lateral brain surface, anterior relative to the IPS (see Figure 1 from Steel et al., 2021 and Figure 3 from Steel et al., 2023). In contrast, PIGS is located on the posterior brain surface, also posterior relative to the IPS. In other words, they are located on two different sides of a major brain sulcus. In this revision we have clarified this point, including the citations by Steel and colleagues in section 4.3.

Comments 2.3: Another reason that it would be helpful to relate PIGS to this scene memory area is that this scene memory area has been shown to have activity related to the amount of visuospatial context (Steel et al, 2023, The Journal of Neuroscience). The conditions used to show the sensitivity of PIGS to ego-motion also differ in the visuospatial context that can be accessed from the stimuli. Even if PIGS appears distinct from the scene memory area, the degree of visuospatial context is an alternative account of what might be represented in PIGS.

Response 2.3: The reviewer raises an interesting point. One minor confusion is that we may be inadvertently referring to two slightly different types of “visuospatial context”. Specifically, the stimuli used in the ego-motion experiment here (i.e. coherently vs. incoherently changing scenes) represent the same scenes, and the only difference between the two conditions is the sequence of images across the experimental blocks. In that sense, the two experimental conditions may be considered to have the same visuospatial “context”. However, it could be also argued that the coherently changing scenes provide more information about the environmental layout. In that case, considering the previous reports that PPA/TPA and RSC/MPA may also be involved in layout encoding (Epstein and Kanwisher 1998; Wolbers et al. 2011), we expected to see more activity within those regions in response to coherently compared incoherently changing scenes. These issues are now more explicitly discussed in the revised article (section 4.6).

**Reviewer 3:**
Comment 3.1: There are few weaknesses in this work. If pressed, I might say that the stimuli depicting ego-motion do not, strictly speaking, depict motion, but only apparent motion between 2s apart photographs. However, this choice was made to equate frame rates and motion contrast between the 'ego-motion' and a control condition, which is a useful and valid approach to the problem. Some choices for visualization of the results might be made differently; for example, outlines of the regions might be shown in more plots for easier comparison of activation locations, but this is a minor issue.

Response 3.1: We thank the reviewer for these constructive suggestions, and we agree with their comment that the ego-motion stimuli are not smooth, even though they were refreshed every 100 ms. However, the stimuli were nevertheless coherent enough to activate areas V6 and MT, two major areas known to respond preferentially to coherent compared to incoherent motion.

**Reviewer #1 (Recommendations For The Authors):**
I enjoyed reading this article. I have a few suggestions for improvement:(1) Delineation from OPA: The OPA has been described in quite similar terms as PIGS, with its involvement in ego-motion (e.g., crawling, walking) and navigation in general (e.g., Dilks' recent work; Bonner and Epstein). The authors address the distinction in section 4.4. Unlike Kamps et al. (2016) and Jones et al. (2023), the authors found weak or no evidence for ego-motion in OPA. They explain this discrepancy with differences in refresh rates and different levels of spatial smoothing of the fMRI data. It is not clear why these fairly small methodological differences would lead to different findings of ego-motion in the OPA. Arguably, the OPA is the closest of the "established" scene areas to PIGS, both in anatomical location and in function. I would therefore appreciate a more detailed discussion of the differences between these two areas.

Response: Jones et al. have also shown that ego-motion TOS/OPA activity when compared to scrambled scenes. This is fundamentally different than what we have shown here, which coherently vs. incoherently changing scenes (i.e. not a small difference). Also, Kamps et al. used static scenes as a control which, considering TOS/OPA motion-selectivity, have a large impact on TOS/OPA response.

(2) Random effects analysis: The authors mention using a "random effects analysis" for several of their experiments. I would ask them to provide more details on what statistical models were used here. Were they purely random-effects models or actually mixed-effects models? What were the factors that entered into the analysis? Providing more detail would make the analysis techniques more transparent.

Response: This point is now clarified in the Methods section.

(3) Data and code availability: The authors write that data and code "are ready to be shared upon request." (section 2.5) In the spirit of transparency and openness, I strongly encourage the authors to make the data publicly available, e.g., on OSF or OpenNeuro. In particular, having probabilistic maps of PIGS available will allow other researchers to include PIGS in their analysis pipelines, making the current work more impactful.

Response: We have made the probabilistic labels available to the public. This point is now highlighted in section 2.5.

(4) Minor comments on the writing that caught my eye while reading the article:Line 27: "in the human brain".

Response: Done.

-Line 30: I don't agree with the characterization of the previous model of scene perception as "simplistic." Adding one additional ROI makes it no less simplistic. Perhaps the authors can rephrase to make this slightly less antagonistic?

Response: Done.

Line 71: it is not clear why NHPs are relevant here.

Response: We decided to keep the text intact.

Line 138" "were randomized".

Response: Done.

Line 152: "consisting".

Response: Done.

Line 155: "sets" (plural).

Response: Done.

Lines 253-255: Why were the 3T spatially smoothed but not the 7T data? This seems odd.

Response: We kept the text intact.

Line 481: "we found strong motion selectivity" (remove "a").

Response: Done.

Line 564: a word is missing, probably: "a stronger effect of ego-motion".

Response: Done.

Line 591: "controlling spatial attention" (remove "the").

Response: Done.

Line 591 and 594: Both sentences start with "However". I think the first of these should not because it is setting up the contrast for the second sentence.

Response: Done.

Line 607: "higher-level" (hyphen).

Response: Done.

Throughout the manuscript: adverbial phrases such as "(in)coherently changing" or "probabilistically localized" do not get a hyphen.

Response: Done.

**Reviewer #2 (Recommendations For The Authors):**
The authors state that "All data, codes and stimuli are ready to be shared upon request". Ideally, these materials should be deposited in appropriate repositories (e.g. OpenMRI, GitHub) and not require readers to contact the authors to obtain such materials.Other Comments:(a) The title ("A previously undescribed scene-selective site is the key to encoding ego-motion in natural environments") is potentially misleading - the work was not conducted in a natural environment. At best, you could say they are 'naturalistic stimuli'. Also, in what sense is PIGS "key" to encoding ego-motion - the study just shows sensitivity to this factor.

Response: We changed the title to “naturalistic environments”.

(b) Figure 1 - I'm not sure what point the authors are trying to make with Figure 1. The comparison is between a highly smoothed, group fixed-effects analysis and a less-smoothed individual subject analysis. The differences between the two could reflect group vs. individual, highly-smoothed (5 mm) versus less-smoothed (2 mm), or differences in thresholding. If the thresholding were lower for the group analysis, it would probably start to look more similar to the individual subject. As it stands, this figure isn't particularly informative, it seems redundant with Figure 2, and Figure 1A is not even referenced in the main text. Further, fixed effects analyses are relatively uncommon in the recent literature, so their inclusion is unusual.

Response: Figure 1A is a replication of the data/method used in Nasr et al., 2011 and it will help the readers see the difference between the “traditional” scene-selectivity maps generated based on group-averaging” vs. data from individual subjects. In this case, we decided not to change the Figure.

(c) Figure 3 - why are the two sets of maps shown at different thresholds? For 3B given the larger sample size, it is expected that the extent of the significant activations will increase. Currently the higher threshold for 3B and the smaller range for 3A is making the sets of maps look more comparable.

Response: As the reviewer noticed, the number of subjects is larger in Figure 3B compared to 3A. The main point of this figure is to show that the PIGS activity center does not vary across populations. Considering this point, we decided not to change this figure.

(d) Figure 10 - why is the threshold lower than used for other figures? It would be helpful if there was consistent thresholding across figures.

Response: Experiment 6 and Experiment 1 are based on different stimuli (see Methods). Also, among those subjects who participated in Experiment 1, two subjects did not participate in Experiment 6. These points are already highlighted in the text.

(e) Figures - how about the AFNI approach of thresholding and showing sub-threshold data at the same time? (Taylor et al, 2023, Neuroimage).

Response: We highly appreciate the methodology suggested by Taylor and colleagues. However, our main point here is to show the center of PIGS activity. In this condition, showing an unthresholded activity map doesn’t have any advantage over the current maps. Considering these points, we decided not to change the figures.

(f) Coherent versus incoherent scenes - there are many differences between the coherent and incoherent scenes. Arguing that it must be ego-motion seems a little premature without further investigation. Activity anterior to OPA has been associated with the construction of an internal representation of a spatial environment (Steel et al., 2023, The Journal of Neuroscience). Could it be that this is the key effect, not really the ego-motion?

Response: In this revision, we discussed the study by Steel et al., 2021 and 2023 in section 4.3.

**Reviewer #3 (Recommendations For The Authors):**
Overall, I think this is already an excellent contribution. The suggestions I have are minor and may help with the clarity of the results.(1) My main request of the authors would be to provide more points of reference in some of the figures with cortical maps. In many cases, the authors use arrows to point to the locations of activations of interest. However, the arrows in adjacent figures are often not placed in exactly the same places on maps that are meant to be compared. It would very much help the viewer to compare activations if the arrows pointing to activations or regions of interest were placed in identical locations for the same brains appearing in different sub-panels (e.g. in panels A and B of Figure 1). The underlying folds of the cortical surface provide some points of reference, but these are often occluded to different extents by data in figures that are meant to be compared.

Response: To address the reviewer’s concern, we regenerated Figure 8 (Figure 7 in the previous submission) and we tried to put arrowheads in identical locations, as much as possible. Especially for PIGS, this point was also considered in Figures 2 and 3.

(2) Outlines (such as those in Figure 5) are also very useful, and I would encourage broader use of them in other figures (e.g. Figures 7, 10, and 12). Figures 10 and 12 are on the fsaverage surface, so the same outlines could be used for them as for Figure 5.To be clear, it's possible to apprehend the results with the figures as they are, but I think a few small changes could help a lot.

Response: In this revision, we added outlines to Figures 11 and 13 (Figure 10 and 12 in the previous submission). We did not add the outline to Figure 8 because it made it hard to see PIGS. Rather we used arrows (see the previous comment).

Other minor points:In the method for Experiment 4, the authors write: "Other details of the experiment were similar to those in Experiment 1.". Similar or the same? The authors should clarify this statement, e.g. "the number of images per block, the number of blocks, the number of runs were the same as Experiment 1" - with any differences noted.

Response: This point is now addressed in the Methods section.

In Figure 8, it would be better to have the panel labels (A, B, C, D) in the upper left of each panel rather than the lower left.

Response: We tried to keep the panels arrangement consistent across the figures. That is why letters are positioned like this.

A final gentle suggestion: pycortex (http://github.com/gallantlab/pycortex) provides a means to visualize the flattened fsaveage surface with outlines for localized regions of interest and overlaid lines for major sulci. Though it is by no means necessary for publication, It would be lovely to see these results on that surface, which is freely available and downloadable via a pycortex command (surface here: https://figshare.com/articles/dataset/fsaverage_subject_for_pycortex/9916166)

Response: We thank the reviewer for bringing pycortex to our attention. We will consider using it in our future studies.